# Functional brain architecture is associated with the rate of tau accumulation in Alzheimer's disease

Nicolai Franzmeier [1]*, Julia Neitzel[1], Anna Rubinski[1], Ruben Smith [2,3], Olof Strandberg[3], Rik Ossenkoppele[3,4], Oskar Hansson [3,5], Michael Ewers[1] & Alzheimer's Disease Neuroimaging Initiative (ADNI)

In Alzheimer's diseases (AD), tau pathology is strongly associated with cognitive decline. Preclinical evidence suggests that tau spreads across connected neurons in an activity-dependent manner. Supporting this, cross-sectional AD studies show that tau deposition patterns resemble functional brain networks. However, whether higher functional connectivity is associated with higher rates of tau accumulation is unclear. Here, we combine resting-state fMRI with longitudinal tau-PET in two independent samples including 53 (ADNI) and 41 (BioFINDER) amyloid-biomarker defined AD subjects and 28 (ADNI) vs. 16 (Bio-FINDER) amyloid-negative healthy controls. In both samples, AD subjects show faster tau accumulation than controls. Second, in AD, higher fMRI-assessed connectivity between 400 regions of interest (ROIs) is associated with correlated tau-PET accumulation in corresponding ROIs. Third, we show that a model including baseline connectivity and tau-PET is associated with future tau-PET accumulation. Together, connectivity is associated with tau spread in AD, supporting the view of transneuronal tau propagation.

[1] Institute for Stroke and Dementia Research, Klinikum der Universitat München, Ludwig-Maximilians-Universitat LMU, Munich, Germany. [2] Department of Neurology, Skane University Hospital, Lund, Sweden. [3] Clinical Memory Research Unit, Department of Clinical Sciences Malmo, Lund University, Lund, Sweden. [4] Alzheimer Center Amsterdam, Department of Neurology, Amsterdam Neuroscience, Vrije Universiteit Amsterdam, Amsterdam UMC, Amsterdam, the Netherlands. [5] Memory Clinic, Skane University Hospital, Malmo, Sweden. A full list of consortium members appears at the end of the paper. *email: nicolai.franzmeier@med.uni-muenchen.de

Alzheimer's disease (AD) is characterized by the hallmark pathologies amyloid-beta (Aβ) and hyperphosphorylated tau, which are associated with neurodegeneration, cognitive decline and dementia[1]. While Aβ forms extracellular plaques that accumulate in a rather diffuse and global manner across the cortex[2], the formation of intracellular neurofibrillary tangles follows a sequential spatio-temporal pattern described by the Braak-stages[3]. Extensive post-mortem findings in AD brains suggest that tau pathology is first confined to the locus coeruleus and entorhinal cortex and subsequently occurs within the allocortex, inferior temporal lobe, association cortices and lastly in primary sensorimotor and visual cortices[4]. The sequential and organized spread of pathologic (i.e. hyperphosphorylated) tau species across anatomically connected brain areas in AD has fostered the idea of a prion-like tau transmission mechanism[5], where neurons carrying pathological tau species transmit pathological tau to their connected neighbors, thereby triggering a self-propagating tau spreading cascade[6,7]. Preclinical studies have shown that pathological tau exhibits prion-like features: First, pathological tau species have been shown to be capable of seeding, i.e. they can serve as templates for misfolding of physiological tau species[8], initiating the formation of pathological tau tangles[9,10]. Second, the induction of hyperphosphorylated tau seeds in circumscribed brain regions of rodents has been shown to elicit spread of pathological tau to anatomically connected regions rather than simple diffusion to spatially adjacent regions[11–13]. In vitro studies have further emphasized that pathological tau spreads specifically across synapses[14] in an activity-dependent manner[15], where greater synaptic connectivity (i.e. shared activity between neurons) facilitates tau spreading both in vitro and in vivo[16]. Thus, higher co-activation of brain regions may facilitate the spreading of tau between functionally connected brain regions.

The combination of recently developed tau-PET imaging with resting-state functional MRI (fMRI) has greatly facilitated efforts to test the relationship between brain connectivity and tau pathology in living AD patients[17]. A recent study combining tau-PET and resting-state fMRI cross-sectionally in a small sample of AD patients found tau deposition to be preferentially distributed within the boundaries of fMRI-detected functional brain networks, i.e. within regions that are functionally connected[18]. We reported previously in cognitively normal elderly individuals and patients with AD or vascular cognitive impairment that higher inter-regional tau covariance (i.e. correlation of tau-PET uptake levels between brain regions) is closely associated with higher fMRI-assessed functional connectivity between the spatially matching brain regions. Furthermore, regions that were highly functionally connected to tau hotspots showed high tau levels, whereas regions with low functional connectivity to tau hotspots harbored little tau[19]. In a similar vein, highly interconnected brain regions—so called hubs—showed more tau PET uptake than less well-connected regions in patients with AD, possibly because their large number of connections increases the likelihood to receive pathological tau species from remote brain regions[20]. Together, these first in vivo findings on the association between tau and brain network architecture support the hypothesis that the accumulation and propagation of pathologic tau is related to neural activity and inter-regional connectivity. However, whether functional connectivity between regions is associated with future tau accumulation rates in connected brain regions in living AD patients is unknown. Understanding the role of functional connectivity in the development and spread of tau pathology is of major clinical importance, paving the way towards targeting neural activity related mechanisms of tau pathology to halt clinical AD progression[21].

Therefore, we assessed in the current study longitudinal AV1451 tau-PET data from 53 amyloid-positive subjects at pre-dementia AD stages and from 28 cognitively normal amyloid-negative (CN Aβ−) control subjects from the ADNI study in combination with baseline resting-state fMRI. For replication of the results, we further included longitudinal tau-PET data from the BioFINDER cohort (i.e. 41 Aβ+ subjects from preclinical to AD dementia stages and 16 CN Aβ−) together with a resting-state fMRI connectome template derived from 500 subjects of the human-connectome project (HCP)[22]. Using pairwise ROI-to-ROI correlation analysis within the AD (i.e. Aβ+) samples, we assessed the covariance in AV1451 longitudinal tau-PET change among 400 brain regions of interest (ROIs) using a standard neocortical parcellation atlas[23]. Based on resting-state fMRI data, we assessed group-mean functional connectivity between the same 400 ROIs. By combining functional connectivity and longitudinal tau-PET data, we pursue our major aim, i.e. we show that the brains' connectivity architecture is associated with the future spread of tau. To this end, we first assessed whether tau deposition at baseline and the rate of tau accumulation are increased in subjects with AD compared to controls. Second, we show that highly functionally connected regions show similar rates of tau accumulation (i.e. covariance in tau change). Third, we emphasize that future tau accumulation can be modeled by combining fMRI-assessed functional brain architecture with baseline tau levels.

## Results

**Sample characteristics.** We included 81 participants from the ADNI3 study (https://clinicaltrials.gov/ct2/show/NCT02854033) all with available baseline resting-state fMRI and longitudinal AV1451 tau-PET data. The sample included 28 CN Aβ− as a healthy reference group, and 32 CN Aβ+ as well as 21 MCI Aβ+ covering the pre-dementia AD spectrum (see Table 1 for baseline demographics). Mean tau-PET follow-up time was $1.3 \pm 0.52$ years in CN Aβ− and, $1.27 \pm 0.46$ years in CN Aβ+ and $1.37 \pm 0.57$ years in MCI Aβ+. No significant differences in tau-PET follow-up time were found across groups ($p = 0.817$, ANOVA). As an independent validation sample, we included 57 subjects from the BioFINDER study with available longitudinal AV1451 tau-PET data (Table 1). This sample included 16 CN Aβ−, 16 CN Aβ+, 7 MCI Aβ+ and 18 subjects with AD dementia (Aβ+). Mean tau-PET follow-up time was $2.03 \pm 0.47$ years in CN Aβ−, $1.91 \pm 0.32$ years in CN Aβ+, $1.82 \pm 0.12$ years in MCI Aβ+ and $1.87 \pm 0.34$ years in AD dementia. For BioFINDER, we used resting-state fMRI data from 500 subjects of the human-connectome project (HCP) to determine a group-mean functional connectivity template that was used for combined tau vs. functional connectivity analyses similar to previous studies[24]. For both samples, all tau-PET images were intensity normalized to the inferior cerebellar grey[25]. Usage of an alternative reference region (i.e. eroded white matter) did not change the currently reported result pattern (data not shown).

**Higher baseline tau- and tau-PET change in Aβ+ vs. CN Aβ−.** First, we assessed baseline and follow-up tau-PET levels within 400 ROIs covering the neocortex[23], as well as longitudinal tau-PET change (i.e. ROI-wise SUVR change per year) for each group and sample. In CN Aβ−, no elevated tau-PET uptake (i.e. surpassing a pre-established tau-PET SUVR threshold > 1.3)[26] was found at baseline or follow-up in both ADNI (Fig. 1a) and Bio-FINDER (Fig. 1d). In CN Aβ+, tau-PET uptake increased across time especially in inferior temporal regions at follow-up in both ADNI (Fig. 1b) and BioFINDER (Fig. 1e), surpassing the threshold for elevated tau-PET of 1.3 in ADNI CN Aβ+ at follow-up (Fig. 1b). In MCI Aβ+, elevated temporal, parietal and frontal tau-PET was found at baseline, with increases at follow-up

**Table 1 Subject characteristics.**

| ADNI | CN-Aβ− (n = 28) | CN-Aβ+ (n = 32) | MCI- Aβ+ (n = 21) | p-value |
|---|---|---|---|---|
| Age | 74.07 (6.82) | 76.88 (5.86) | 74.86 (7.57) | 0.251 |
| Sex (m/f) | 12/16 | 12/20 | 11/10 | 0.564 |
| Education (M/SD) | 16.8 (1.94) | 15.0 (4.56) | 15.62 (2.92) | 0.214 |
| MMSE (M/SD) | 29.25 (1.02)[c] | 28.5 (2.13)[c] | 26.53 (4.03)[a,b] | 0.010 |
| ADAS-global (M/SD) | 7.93 (4.27)[c] | 12.42 (7.16)[c] | 17.61 (10.38)[a,b] | 0.001 |
| ApoE ε4 status (pos/neg) | 5/23 | 19/13 | 14/7 | 0.002 |
| Global AV45 SUVR(M/SD) | 1.04 (0.04)[b,c] | 1.33 (0.21)[a] | 1.34 (0.13)[a] | <0.001 |
| Mean tau-PET follow-up time in years (M/SD) | 1.30 (0.52) | 1.27 (0.46) | 1.37 (0.57) | 0.817 |

| BioFINDER | CN-Aβ− (n = 16) | CN-Aβ+ (n = 16) | MCI- Aβ+ (n = 7) | AD dementia (n = 18) | |
|---|---|---|---|---|---|
| Age | 73.88 (5.32) | 75.44 (6.09) | 72.71 (6.63) | 69.83 (10.48) | 0.192 |
| Sex (m/f) | 10/6 | 6/10 | 2/5 | 11/7 | 0.245 |
| Education (M/SD) | 12.59 (4.06) | 10.56 (3.22) | 11.14 (2.67) | 13.44 (3.26) | 0.097 |
| MMSE (M/SD) | 29 (1.1)[d] | 29.31 (1.08)[d] | 25.57 (2.94) | 22.06 (5.17)[a,b] | <0.001 |
| ADAS-delayed recall (M/SD) | 1.81 (1.47)[c,d] | 2.31 (1.49)[c,d] | 6.17 (2.4) | 7.62 (2.45)[a,b] | <0.001 |
| ApoE ε4 status (pos/neg) | 0/16 | 10/6 | 4/3 | 11/7 | <0.001 |
| Global Flutemetamol SUVR | 0.52 (0.03)[b,c,d] | 0.77 (0.12)[a,d] | 0.84 (0.14)[a] | 0.97 (0.15)[a,b] | <0.001 |
| Mean tau-PET follow-up time in years (M/SD) | 2.03 (0.47) | 1.91 (0.32) | 1.82 (0.12) | 1.97 (0.34) | 0.484 |

p-values were derived from ANOVA for continuous measures and from Chi-squared tests for categorical measures
*M* male, *f* female, *MMSE* Mini-Mental State Exam, *ADAS* Alzheimer's disease assessment scale, cognitive subscale
Mean values significantly (p < 0.05) different from—
[a]CN-Aβ−
[b]CN-Aβ+
[c]MCI-Aβ+
[d]AD dementia

(ADNI: Fig. 1c; BioFINDER: Fig. 1f). A spatially similar longitudinal tau-PET increase was found in AD dementia subjects of the BioFINDER sample (Fig. 1g). In CN Aβ−, ROI-wise *t*-tests on the longitudinal tau-PET change maps confirmed that there was no (ADNI: Fig. 1h) or only minor (BioFINDER: Fig. 1j) significant tau-PET increase from baseline to follow-up. In contrast, the Aβ+ groups showed significant and widespread tau accumulation (i.e. ROI-wise *t*-test, p < 0.005) across temporal, parietal and frontal regions in ADNI (Fig. 1i) and BioFINDER (Fig. 1k). These analyses support the notion that AD (i.e. Aβ+) subjects show elevated tau levels and faster tau accumulation than controls.

**Functional connectivity and covariance in tau accumulation**. We assessed whether regions with strong functional connectivity show higher covariance in tau pathology accumulation, indicative of transneuronal tau spread. To this end, we tested the association between resting-state fMRI functional connectivity between two given brain regions and the covariance in annual tau-PET change between the same brain regions. This main analysis was conducted in the AD groups (i.e. 53 Aβ+ subjects in ADNI and 41 Aβ+ subjects in BioFINDER), since there were no (ADNI) or only minor (BioFINDER) longitudinal tau-PET changes in the CN Aβ− groups (see Fig. 1d). For ADNI, subject-specific functional connectivity matrices were obtained on preprocessed resting-state fMRI data using a standardized functional brain atlas including 400 ROIs[23] (Fig. 2a) and subsequently averaged to a group-mean connectivity matrix (Fig. 2b). For BioFINDER, we obtained a $400 \times 400$ template of functional connectivity from preprocessed resting-state fMRI data from 500 HCP subjects (Fig. 2b). Within both ADNI and BioFINDER, we determined for each Aβ+ subject the annual tau-PET change within each of the 400 ROIs. We then computed the across-subject Spearman correlation in tau change, yielding a $400 \times 400$ matrix of covariance in tau change (Methods illustrated in Fig. 3a), where higher correlations indicate similar annual tau change rates within a given ROI pair (Fig. 3b). To test whether functionally connected

regions show similar tau accumulation rates, we performed spatial regression using the group-average functional connectivity matrices as a predictor of the covariance in tau change matrix (Fig. 3b). Supporting our hypothesis, we found a positive association between functional connectivity and covariance in tau change in both ADNI (β = 0.38, p < 0.001, see Fig. 4a) and BioFINDER (β = 0.30, p < 0.001, Fig. 4b). We repeated the same analyses 200 times generating in each trial a new connectivity null-model (i.e. shuffled connectivity matrix with preserved degree- and weight-distribution), yielding β-value distributions of M/SD = 0.09/0.002 for ADNI and M/SD = −0.02/0.002 for BioFINDER. Comparing the β-values estimated in the ADNI and BioFINDER samples with these null-distributions using an exact test (i.e. the percentage of null-distribution derived β-values surpassing the true β-value), yielded p-values < 0.001 for both ADNI and BioFINDER. These analyses support the notion that higher functional connectivity is associated with more similar annual tau accumulation rates in AD. Importantly, the association between functional connectivity and covariance in tau-PET change remained consistent when controlling for the Euclidean distance between each ROI pair (ADNI: β = 0.30, p < 0.001, BioFINDER: β = 0.22, p < 0.001), or when controlling the assessment of covariance in tau change for age, sex, education, ApoE4 status, diagnosis and MMSE (ADNI: β = 0.38, p < 0.001, BioFINDER: β = 0.24, p < 0.001). These findings support the view that tau spread is predominantly determined by connectivity rather than proximity[13], and independent of age, sex, education or ApoE. When exploratorily testing the association between functional connectivity and covariance in tau-PET change in the Aβ− control subjects of the BioFINDER cohort, who showed minor tau accumulation (Fig. 1), we found a less strong but congruent positive association between functional connectivity and covariance in tau-PET change (β = 0.10, p < 0.001), which remained after controlling for Euclidean distance (β = 0.09, p < 0.001).

To exploratorily test whether the association between functional connectivity and covariance in tau-PET change was higher in Aβ+ than in Aβ−, we iteratively determined covariance in

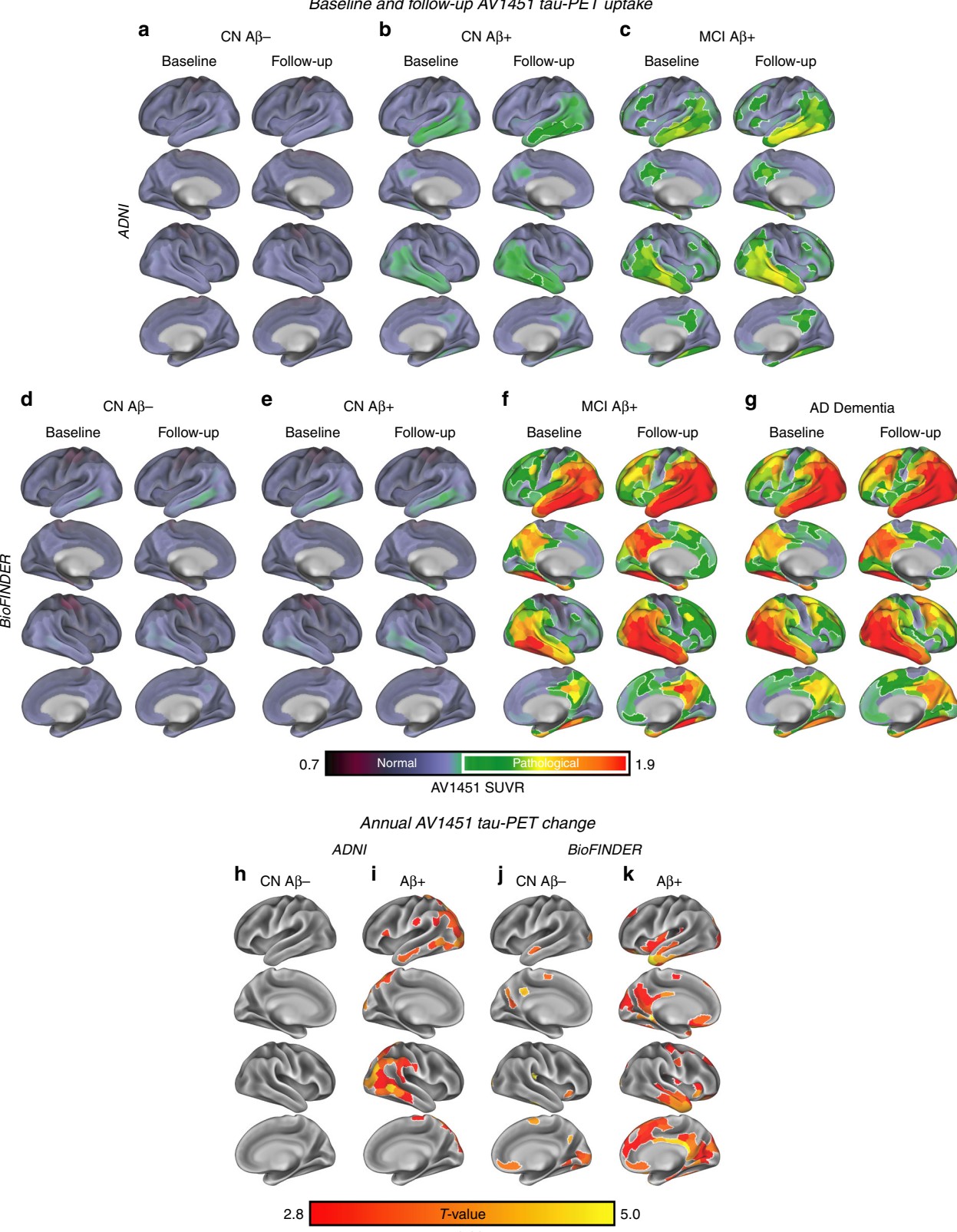

**Fig. 1 Group-average tau-PET levels.** Group-average tau-PET levels for the ADNI (**a–c**) and BioFINDER (**d–e**) samples, stratified by diagnostic group. Tau-PET levels are showed as continuous values, where pathological tau-PET levels (i.e., surpassing an SUVR of 1.3)[26] are highlighted by white outlines in Panels **a**–**g**. Group-average data are shown in ADNI for 28 CN Aβ− (**a**), 32 CN Aβ+ (**b**) and 21 MCI Aβ+ (**c**) at baseline and follow-up. For BioFINDER, group-average data are shown for 16 CN Aβ− (**d**), 16 CN Aβ+ (**e**), 7 MCI Aβ+ (**f**) and 18 AD Dementia patients (**g**). In ADNI, ROI-wise t-tests against zero show no significant annual tau-PET increases in 28 CN Aβ− (**h**), but significant (p < 0.005) temporo-parietal tau-PET changes in 53 Aβ+ (**i**). In BioFINDER, t-tests against zero show minor annual tau-PET increases in the 16 CN Aβ− (**j**), but widespread temporal, parietal and frontal tau-PET increases in the 41 Aβ+ subjects (**k**).

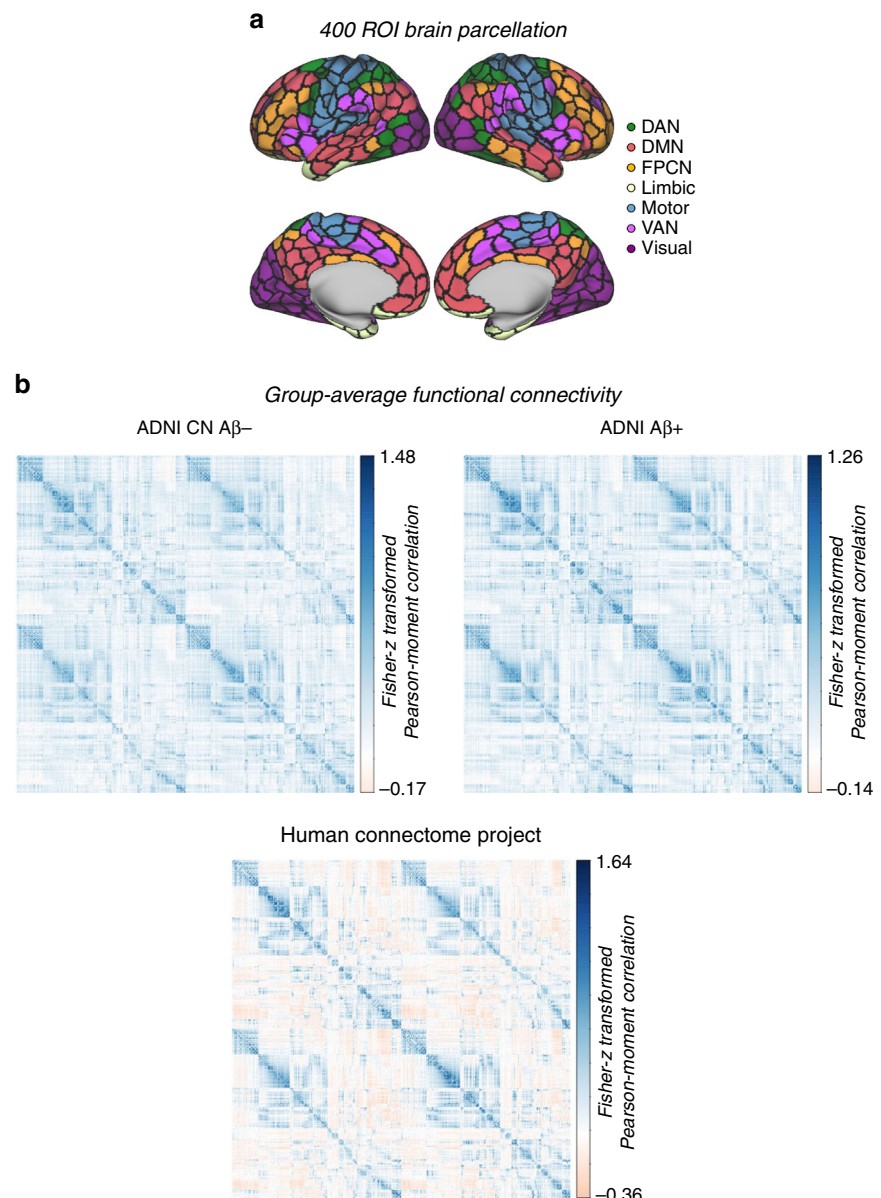

**Fig. 2 Brain parcellation and resting-state fMRI. a** Surface rendering of the 400 ROI brain parcellation that was applied to tau-PET and resting-state fMRI data for ROI based analyses. **b** Group-average functional connectivity matrices for 28 CN Aβ− and 53 Aβ+ of the ADNI sample, as well as for 500 subjects from the human-connectome project (HCP). In ADNI, no Bonferroni-corrected differences ($p < 0.05$) in functional connectivity were found between the Aβ+ and CN-Aβ− in ANCOVAS controlling for age, sex, education and diagnosis.

tau-PET change in BioFINDER using 1000 bootstrapping iterations (i.e. random sampling from the subject pool with replacement), based on which we re-assessed the association between functional connectivity and bootstrapped covariance in tau change for each of the 1000 iterations. The resulting distribution of β-values (i.e. reflecting the association between functional connectivity and covariance in tau change) was significantly higher in Aβ+ (mean/SD = 0.22/0.03) than in Aβ− (mean/SD = 0.06/0.02) as indicated by a two-sample $t$-test ($t(1998) = 145.11$, $p < 0.001$).

Due to the availability of both subject-level tau-PET and functional connectivity data in ADNI, we further assessed the within-sample robustness of the association between functional connectivity and covariance in tau change, by repeating the above described analysis on 1000 bootstrapped samples in the ADNI Aβ+ group. Specifically, we drew 1000 random samples with replacement from the entire pool of 53 Aβ+ subjects and

determined within each bootstrapped sample the group-average functional connectivity and covariance in tau change. For each bootstrapped sample, we tested the association between functional connectivity and covariance in tau change as described above and saved the resulting β-values. The distribution of the 1000 resulting β-values (i.e. regression-derived association between functional connectivity and covariance in tau change), was significantly greater than zero ($t(999) = 238.93$, $p < 0.001$), where the 95% confidence interval did not include zero (95% CI = [0.232; 0.360]). This suggests a robust association between functional connectivity and covariance in tau change within the ADNI sample. Further, we repeated the above described analyses by thresholding the ROIs to a subset with highest tau-PET uptake at baseline (25/50/75% percentile). Here, we found that functional connectivity was associated with higher covariance in tau-PET change across different percentile thresholds of tau-PET (ADNI: 25/50/75%, β = 0.43/0.44/0.48, all $p < 0.001$, BioFINDER: 25/50/

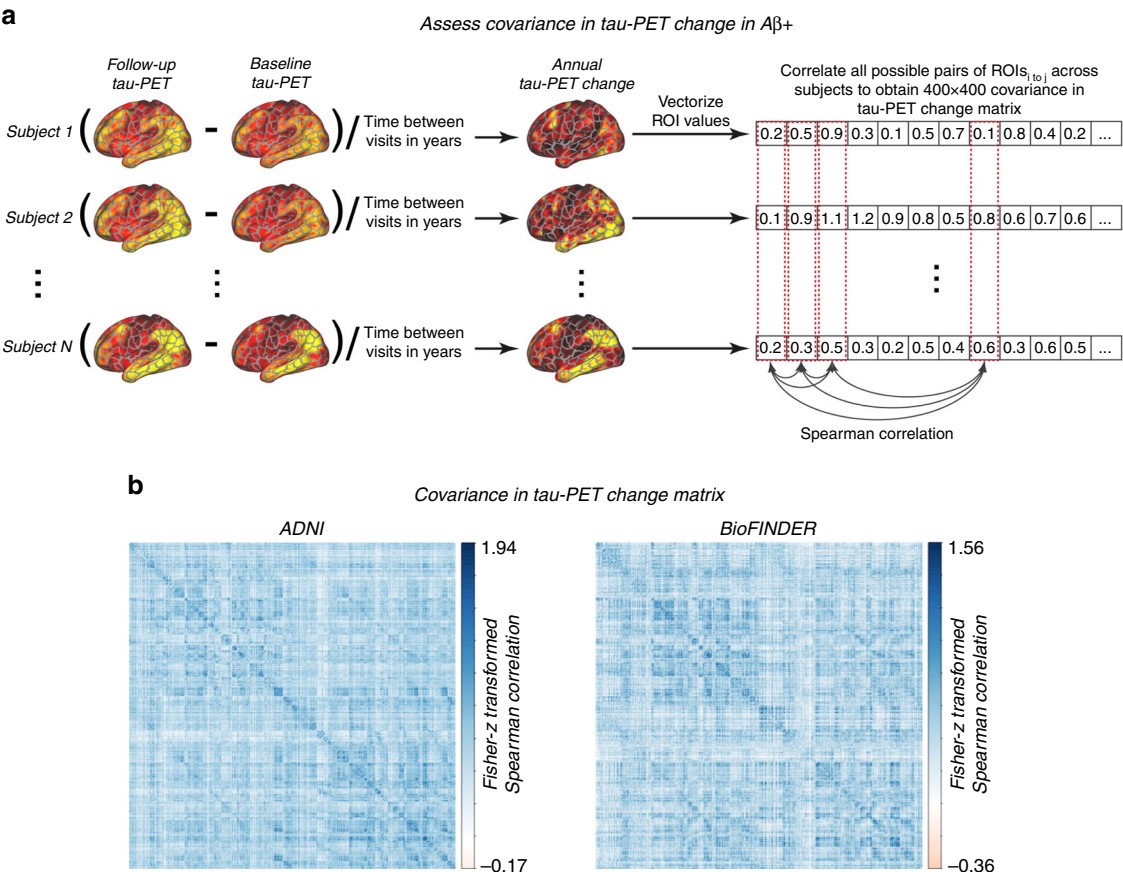

**Fig. 3 Assessment of covariance in tau-PET change. a** Assessment of covariance in tau-PET change. In a first step, annual change in tau-PET was determined as the ROI-wise difference in tau-PET between baseline and follow-up divided by the time between both tau-PET assessments in years. ROI-specific annual tau-PET change scores were vectorized for each of the Aβ+ subjects within each sample, yielding subject-specific 400-element vectors. Within the Aβ+ groups of the ADNI ($N = 53$) and BioFINDER ($N = 41$) samples, the subject-specific 400-element vectors were concatenated across subjects to a 53 × 400 (ADNI) and 41 × 400 (BioFINDER) matrix, where we assessed the Spearman correlation in tau-PET change between ROIs across subjects, yielding a (**b**) 400 × 400 covariance in tau-PET change matrix for each sample that was subsequently Fisher-z transformed.

75%, β = 0.34/0.36/0.39, all $p < 0.001$), where this association became stronger at higher tau-PET percentile thresholds. Lastly, we tested whether the association between functional connectivity and covariance in tau-PET change was consistent across all seven canonical networks shown in Fig. 2a. Here, we found significant positive β-values (ranging from 0.46 to 0.64 in ADNI, Fig. 4c, and 0.18 to 0.50 in BioFINDER, Fig. 4d) between functional connectivity and covariance in tau-PET change within all seven networks. To further illustrate the association between functional connectivity and tau accumulation we mapped the group-mean annual tau change for each ROI in the context of the functional connectome for each sample (Fig. 5).

**Regional tau and tau accumulation in connected regions**. In a follow-up step, we extended this analysis by testing whether the level of tau-PET change in a given seed ROI is associated with the tau-PET changes in closely connected regions in Aβ+. This analysis is based on the idea that if tau spreads as a function of functional connectivity, then interconnected regions should show similar tau accumulation rates. To test this hypothesis, we rank-ordered all ROIs according to their level of annual tau-PET change. We reasoned that at high levels of tau-PET change in seed regions, higher seed FC should be associated with higher tau-PET change in target regions. Vice versa, at lower levels of tau-PET change in the seed regions, higher seed FC should be associated with lower levels of tau-PET change in the target region.

Using linear regression, we tested for each rank-ordered ROI (seed) within each sample, the group-average FC strength to the remaining ROIs (target) as a predictor of the group-average level of tau-PET change in the target ROIs (methods illustrated in Fig. 6a). As hypothesized, we consistently found in both ADNI and BioFINDER that depending on the level of tau-PET change in the seed region, the predictive value of FC for the level of tau change in the target region changed. Specifically, for seeds with high tau accumulation rates, higher FC was associated with higher tau accumulation rates in target regions (i.e. positive β-values in the regression). Conversely, for seeds with low-tau accumulation rates, higher FC was associated with lower tau accumulation rates in target regions (i.e. negative β-values in the regression). This result pattern was mirrored in a strong positive association between the seed ROIs' annual tau-PET change rate and their FCs' predictive weight (i.e. regression-derived β-value) on annual tau-PET change in target ROIs in both the ADNI (Fig. 6b, β = 0.757, $p < 0.001$) and BioFINDER sample (Fig. 6e, β = 0.603, $p < 0.001$). The same analyses applied to 200 null-model connectomes, yielded β-distributions of M/SD = 0/0.07 for ADNI and M/SD = 0.05/0.05 for BioFINDER, yielding exact p-values of $p < 0.001$ when comparing the true β-values against the null-distributions. Again, this analysis was repeated in ADNI using 1000 bootstrapped samples using the above described bootstrapping approach, where the resulting distribution of β-coefficients was significantly different from zero (95% CI =

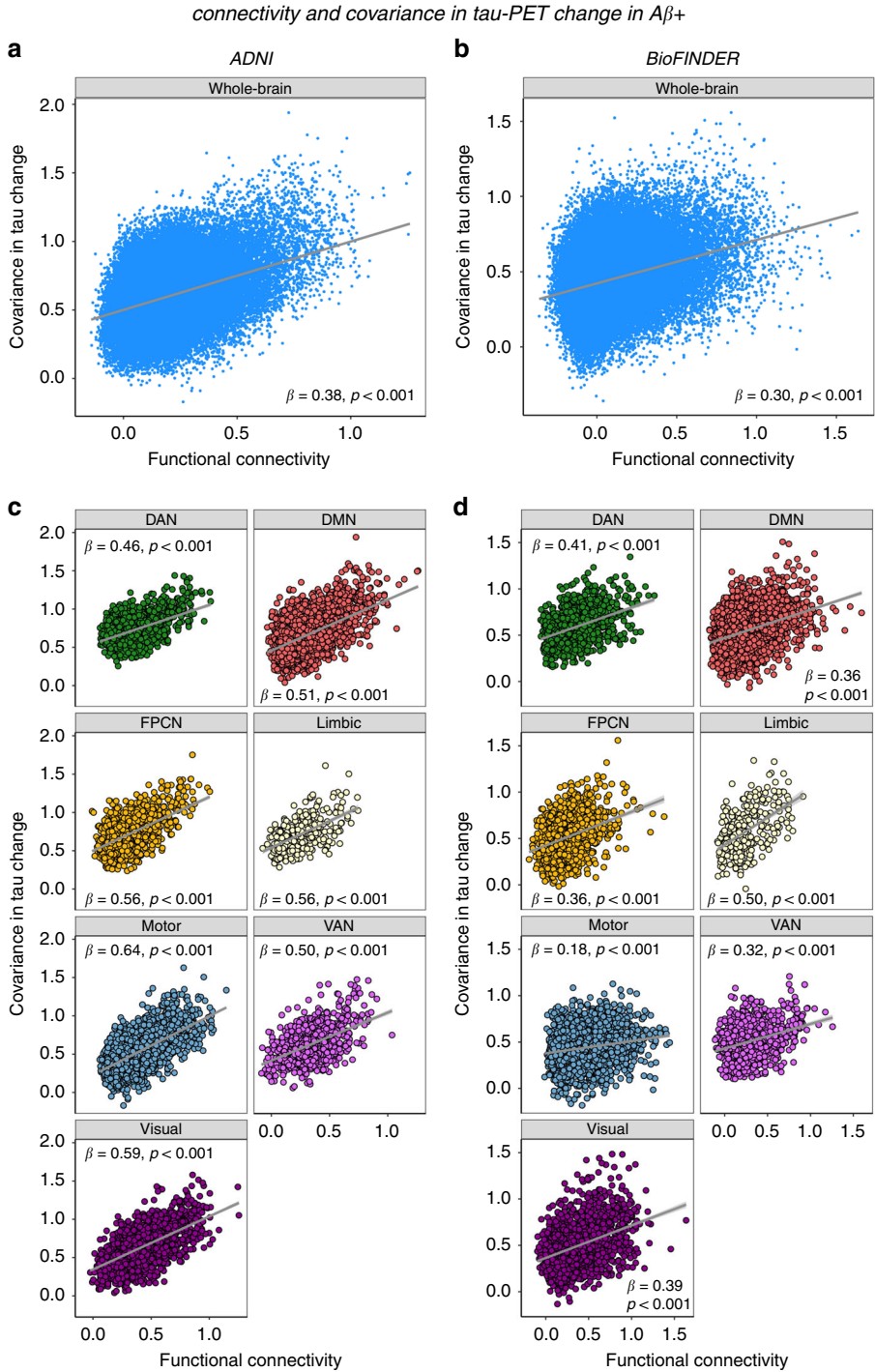

**Fig. 4 Association between functional connectivity and covariance in tau-PET change.** Scatterplots illustrating the association between group-average functional connectivity and covariance in tau-PET change in the Aβ+ groups of the ADNI ($N = 53$) and BioFINDER ($N = 41$) samples, for the whole brain (**a**, **b**) or for the seven canonical brain networks (**c**, **d**). Standardized β- and p-values were derived from linear regression. Source data are provided in a Source data file.

[0.660;0.769], t(999) = 655.25, $p < 0.001$). To illustrate this point further we display the results for the analysis of seed regions with peak-level (Fig. 6c for ADNI and 6f for BioFINDER) and minimum-level of tau-PET change (Fig. 6d for ADNI and 6g for BioFINDER).

**Group-level spreading models of tau-PET change.** For our major aim, we model tau spreading based on baseline tau levels,

functional network architecture and spatial remoteness of connections. In other words, we tested whether tau spreads from affected regions preferentially along functionally strong and spatially short connections (illustrated in Fig. 7a). To test this hypothesis in a systematic way, we employed three approaches: As a first negative control, we tested whether tau spread can be simply modeled by the tau pattern at baseline and the Euclidean distance between ROIs. The rationale for this negative control is that tau

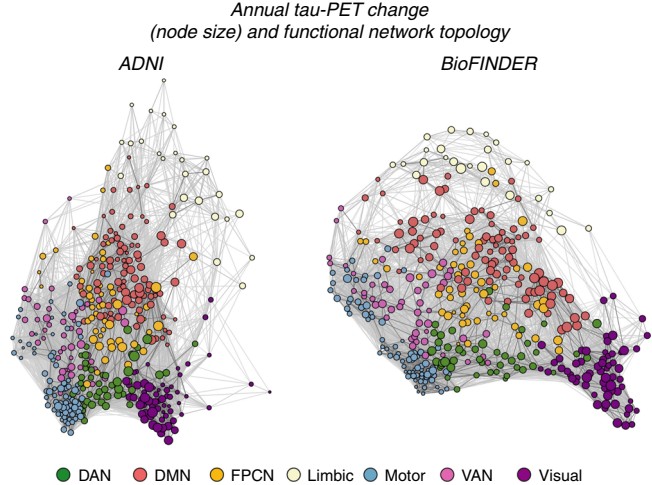

Annual tau-PET change
(node size) and functional network topology

ADNI                              BioFINDER

● DAN  ● DMN  ● FPCN  ○ Limbic  ● Motor  ● VAN  ● Visual

**Fig. 5 Tau-PET change and functional network topology.** Force-directed graphs illustrating ROI-specific annual tau-PET change in the Aβ+ (node size) subjects (ADNI, $N = 53$, left panel; BioFINDER, $N = 41$, right panel) within the context of functional connectivity (node distance, defined based on the Fruchtermann-Reingold algorithm applied to group-average functional connectivity data).

spreads in a diffusive manner to spatially adjacent regions (i.e. regions with a shorter Euclidean distance) independent of functional connectivity (Fig. 7a, Model 1). Second, we tested whether tau spread can be better modeled by combining tau at baseline and functional connectivity. To this end, we assessed tau-weighted functional connectivity for each ROI, that is we determined the mean functional connectivity between a given tau-receiving target ROI and all other tau-seeding 399 seed ROIs (i.e. weighted degree), after weighting each connectivity value by the respective seed ROIs baseline tau level (Fig. 7a, Model 2). Here, strong connections with high tau in the seed ROI have a high weight as they are assumed to explain more variance in future tau accumulation in a given target ROI. In contrast, strong connections with low tau in the seed region will be assigned a low weight, as they are hypothesized to contribute little to explaining future tau accumulation in a connected target ROI. Third, we tested whether this model could be further improved by including the Euclidean distance between each ROI pair as an additional weighting factor, since we reasoned that tau spread should be fastest across functionally strong and spatially short connections (Fig. 7a, Model 3).

Each approach yielded a 400-element vector for the ADNI and BioFINDER sample representing (1) tau-weighted by distance, (2) tau-weighted functional connectivity and (3) tau- & distance-weighted functional connectivity scores for each ROI (see also the surface renderings of the predicted tau change in Fig. 8). Using linear regression, we then tested whether the resulting 400-element vectors were associated with actual annual tau-PET changes in the corresponding 400 ROIs. For analyses including functional connectivity, we again report regular $p$-values derived from linear regression as well as exact $p$-values derived from comparing the linear-regression-derived β-value with the null-distribution β-values assessed on 200 null-model connectomes. For our negative control (Fig. 7a, Model 1: distance-weighted tau), we found a moderate association with future tau change in ADNI (β = 0.252, $p < 0.001$, $R^2 = 0.064$, Fig. 7b), where shorter distance was associated with higher future tau spread. However, no significant association between distance-weighted tau and tau change was found for BioFINDER (β = 0.046, $p = 0.364$, $R^2 = 0.002$, Fig. 7e). For tau-weighted functional connectivity as predictor (Fig. 7a, Model 2) we found a significant effect in ADNI

(β = 0.471, $p < 0.001$, $R^2 = 0.222$, Fig. 7c, β-value null-distribution M/SD = 0.08/0.02, $p_{exact} < 0.001$) and BioFINDER (β = 0.400, $p < 0.001$, $R^2 = 0.160$, Fig. 7f, β-value null-distribution M/SD = 0.04/0.03, $p_{exact} < 0.001$), where higher tau-weighted functional connectivity was associated with faster longitudinal tau change. The strength of this association could be further improved by including Euclidean distance between ROIs as an additional weighting factor (Fig. 7a, Model 3: ADNI: β = 0.499, $p < 0.001$, $R^2 = 0.249$, Fig. 7d, β-value null-distribution M/SD = 0.16/0.02, $p_{exact} < 0.001$; BioFINDER: β = 0.421, $p < 0.001$, $R^2 = 0.177$, Fig. 7g, β-value null-distribution M/SD = 0.02/0.02, $p_{exact} < 0.001$). In ADNI, we again assessed the within-sample robustness of these associations, by computing bootstrapped β-value distributions between annual tau-PET change (dependent variable) and each of the three models (distance-weighted tau, tau-weighted functional connectivity and tau- and distance-weighted functional connectivity), that were derived on 1000 randomly drawn (i.e. bootstrapped) samples. When comparing the bootstrapped β-distributions (Fig. 7h) via ANOVA, we found the highest values for tau- and distance-weighted functional connectivity (Model 3), followed by tau-weighted functional connectivity (Model 2) and lastly distance-weighted tau (Model 1, negative control).

**Subject-level spreading models of tau-PET change.** Using subject-level functional connectivity and tau-PET data in ADNI Aβ+, we repeated the above described analyses, testing for each individual whether (1) distance-weighted tau, (2) tau-weighted functional connectivity and (3) tau- and distance-weighted functional connectivity were associated with individual annual tau change rates (see Fig. 7i). In line with our group-level results, we found on average the strongest association for tau spreading (i.e. regression-derived β-values) using tau- and distance-weighted functional connectivity (Model 3), followed by tau-weighted functional connectivity (Model 2) and lastly distance-weighted tau (Model 1, negative control). β-values of Model 1 were lower than those of Model 2 ($p < 0.001$) and Model 3 ($p < 0.001$) as shown by an ANOVA. In BioFINDER, we conducted the same analyses, using subject-level tau-PET data together with the HCP-derived mean functional connectivity matrix. Here, we found a congruent result pattern (Fig. 7j), where prediction performance was highest for tau- and distance-weighted functional connectivity (Model 3), followed by tau-weighted functional connectivity (Model 2) and distance-weighted tau (Model 1, negative control). β-values of Model 1 were lower than those of Model 2 ($p < 0.001$) and Model 3 ($p < 0.001$), and β-values of Model 2 were lower than those of Model 3 ($p < 0.005$) as shown by an ANOVA. These results support the idea that the regional tau accumulation rates are most strongly associated with the level of tau in connected regions while also taking into account the approximate length of the connection. Lastly, we tested whether the association strength (i.e. β-values) was influenced by age, sex or ApoE status. Using ANOVAs for sex and ApoE and linear regression for age, we found no significant effect ($p > 0.05$) of either measure on the β-value distributions of Models 1–3 in both ADNI and BioFINDER, suggesting age, sex and ApoE do not influence these associations.

## Discussion

The major aim of the current study was to test whether functional connectivity is associated with future tau accumulation in AD. In two independent samples, we found that AD subjects show stronger tau accumulation than amyloid-negative control subjects, and that functionally connected regions show correlated tau accumulation rates. Specifically, regions with high-tau

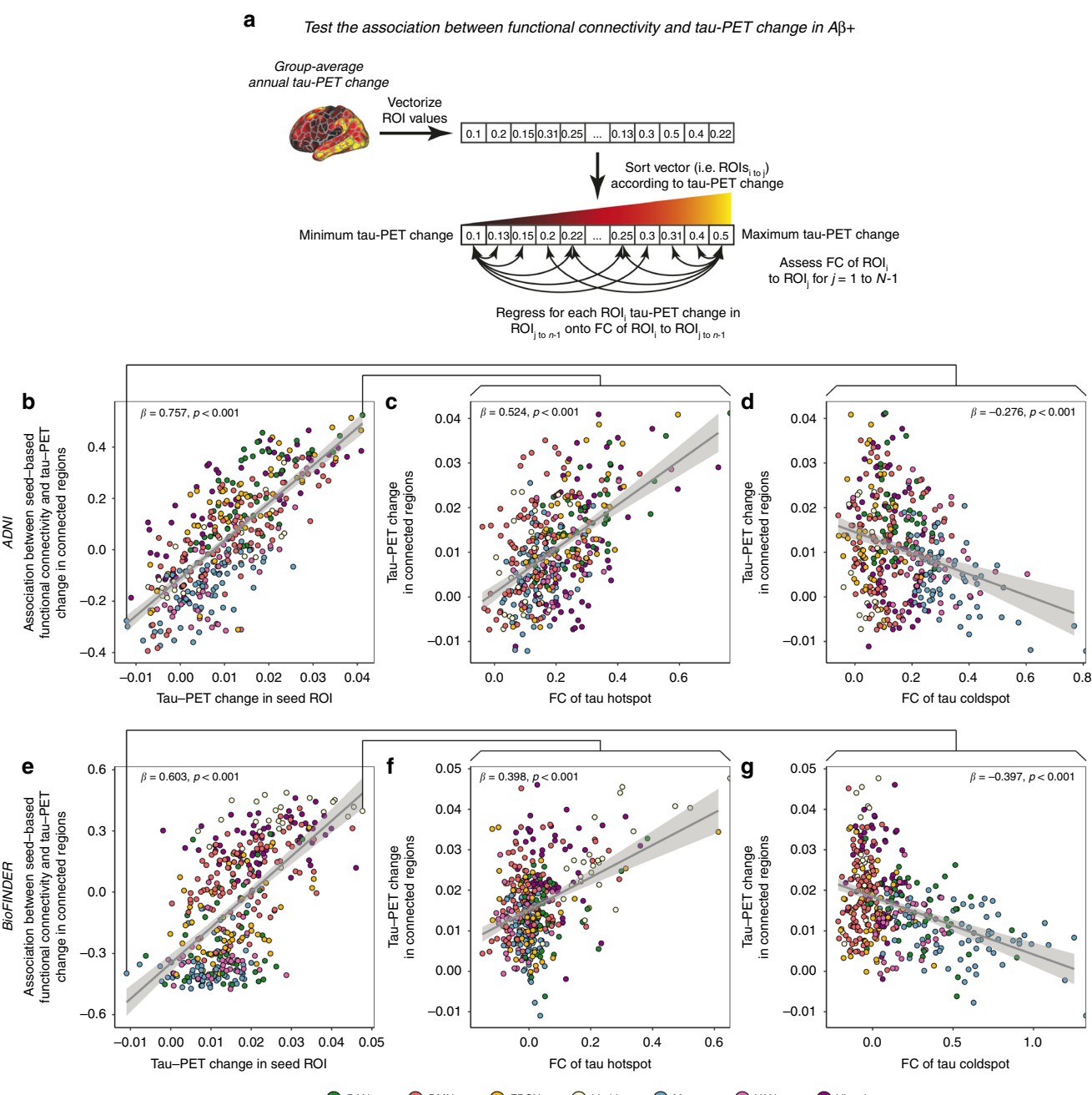

**Fig. 6 Associations between functional connectivity and tau-PET change. a** Pipeline for testing the association between group-average functional connectivity and annual tau-PET change in the 53 Aβ+ from ADNI and 41 Aβ+ subjects from BioFINDER. subjects. For both ADNI (**b**) and BioFINDER (**e**), we plotted the association between annual tau-PET change of a seed-ROI (*x*-axis) and the regression-derived association between its' functional connectivity to target regions and tau-PET change in the respective target regions (*y*-axis). Positive y-values indicate that higher FC to target regions is associated with higher annual tau-PET change, while negative y-values indicate that higher FC to target regions is associated with lower annual tau-PET changes. Illustration of the association between seed-based functional connectivity (*x*-axis) and annual tau-PET change in connected regions (*y*-axis) for ROIs with maximum (ADNI: c; BioFINDER: f) and minimum (ADNI: d; BioFINDER: g) annual tau-PET change. Linear model fits are indicated together with 95% confidence intervals. Source data are provided in a Source data file.

accumulation rates were preferentially connected to other regions with high-tau accumulation rates, whereas regions with low-tau accumulation rates showed high connectivity to regions with similarly low-tau accumulation rates. Most importantly, we found that future tau accumulation in AD could be modeled by a combination of baseline tau levels, functional connectivity and distance between brain regions. Together, our results support the view of pathological tau transmission across neuronal connections in AD.

For our first finding, we show significant tau accumulation in AD subjects that is mostly absent in amyloid-negative controls. This finding is a core prediction of the amyloid cascade model[2] and is in line with previous evidence, showing that tau accumulation is accelerated in the presence of elevated amyloid levels[27].

Second, we could show that functionally highly connected regions show similar tau accumulation rates in AD. This finding critically extends previous results, showing that cross-sectionally assessed tau levels covary between functionally connected regions

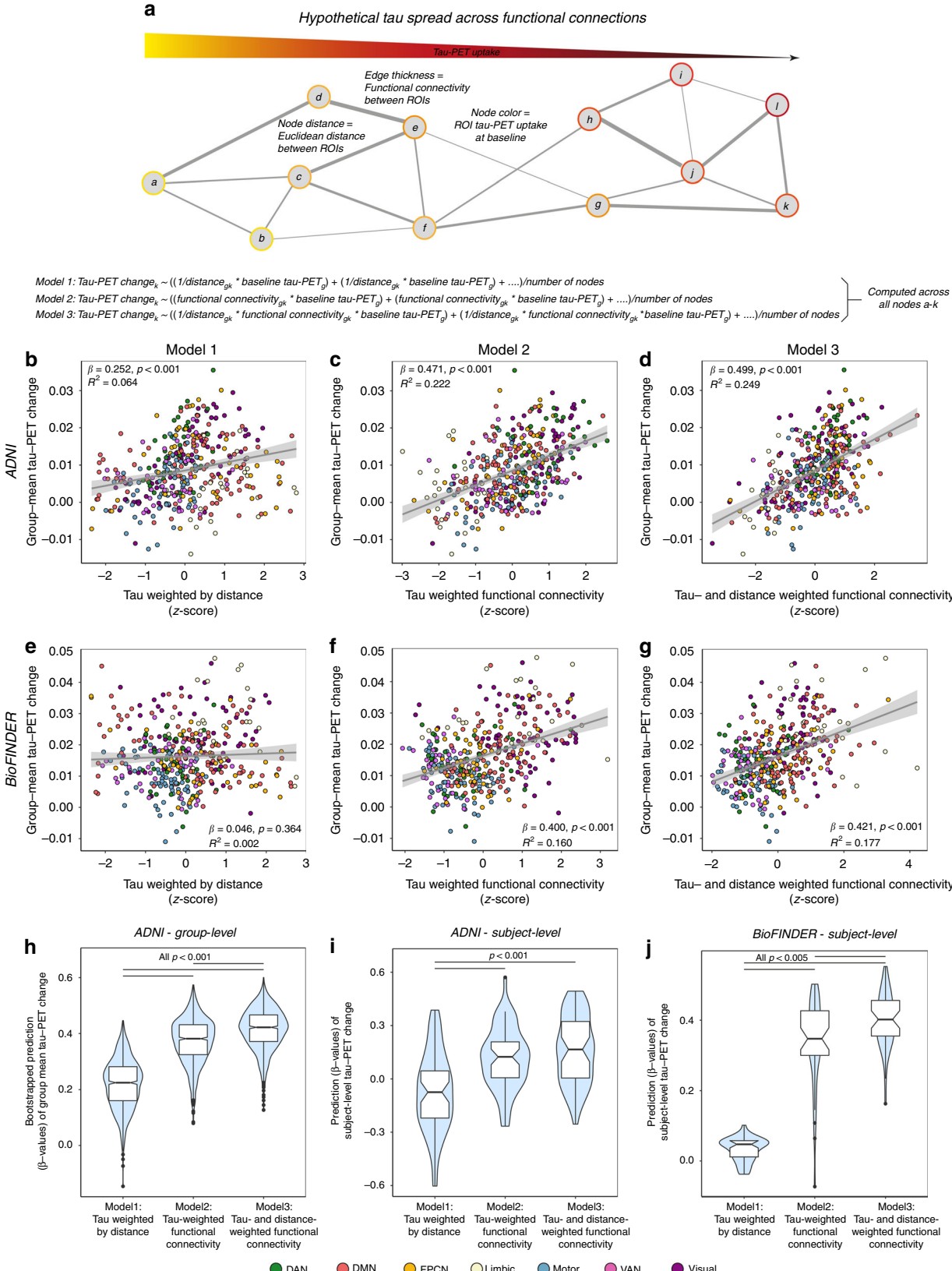

in AD[18,19], and that similar tau levels are found within circumscribed functional networks[28]. In a similar vein, others could show that functional hubs—i.e. brain regions that are highly interconnected with the rest of the brain—harbor high-tau levels, potentially since their large number of connections increases the

likelihood to receive pathological tau species from other regions[20]. While previous cross-sectional findings provided first evidence that tau deposition patterns are related to functional brain architecture, our results critically substantiate these findings, showing that higher functional connectivity between brain regions

**Fig. 7 Prediction of longitudinal tau-PET change. a** Hypothetical network spreading model of tau pathology. Each node within the network represents a brain region, where color indicates local tau pathology, distance between regions indicates connection length (i.e. Euclidean distance) and edge thickness indicates functional connectivity strength. Example formulas for models 1–3 illustrate how we computed tau-weighted distance (Model 1), tau-weighted functional connectivity (Model 2) or tau- & distance-weighted functional connectivity (Model 3) that were used to model group-mean annual tau-PET change in the 53 Aβ + ADNI (**b**–**d**) and 41 Aβ + BioFINDER subjects (**e**–**g**). For ADNI, we computed the association illustrated in (**b**–**d**) for 1000 bootstrapped samples (**h**). Resulting β-value distributions (y-axis) were compared between Models 1–3 using an ANOVA with post-hoc Tukey-test (x-axis). **f** Prediction models 1–3 were assessed on the subject-level for 53 ADNI Aβ+ and 41 BioFINDER Aβ+ subjects using subject-level annual tau-PET change and subject-level connectivity (ADNI) or HCP-derived group-level functional connectivity (BioFINDER). Subject-derived β-value distributions were compared across Models 1–3 using an ANOVA. Source data are provided in a Source data file. Linear model fits are indicated together with 95% confidence intervals.

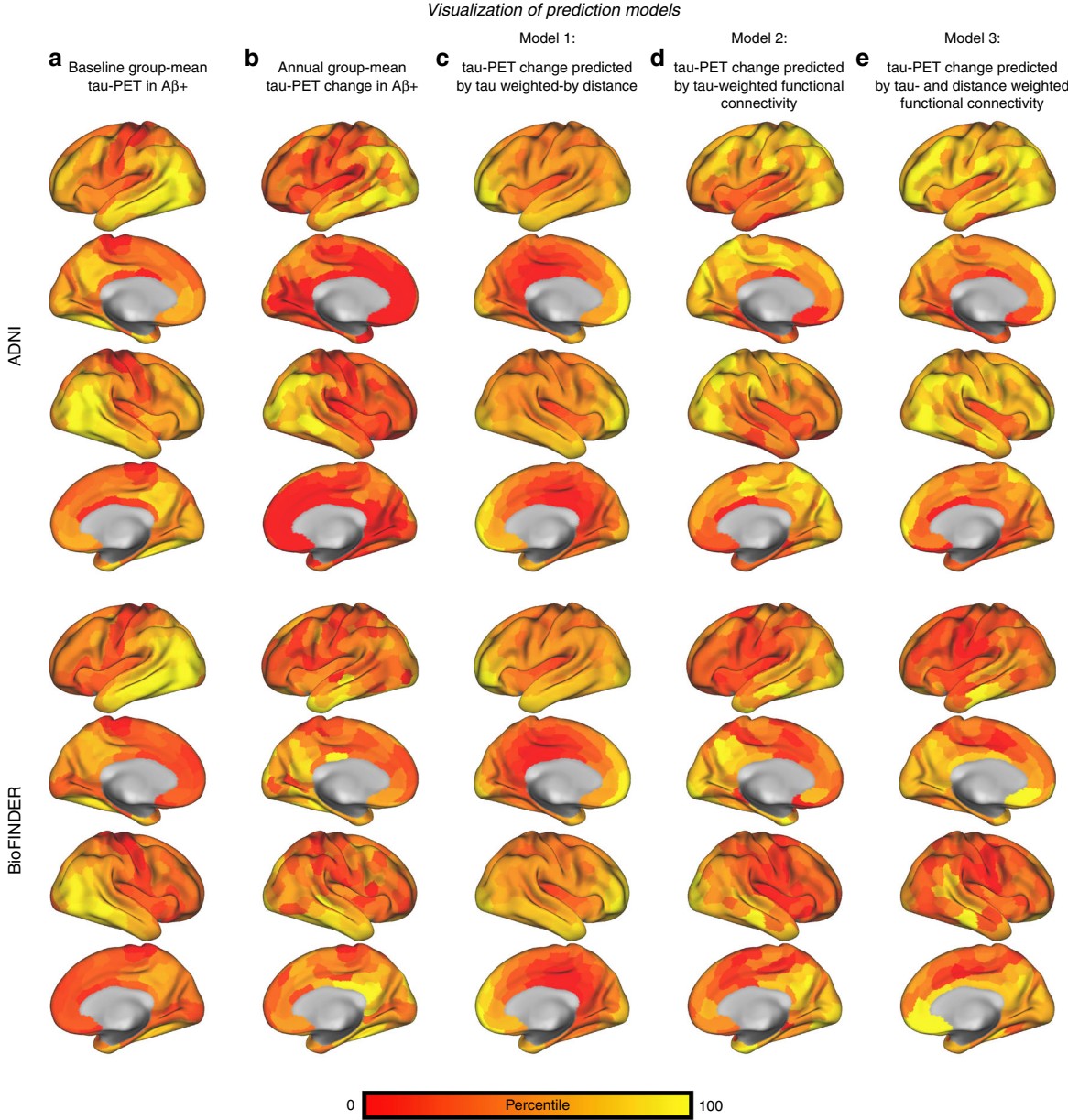

*Visualization of prediction models*

**a** Baseline group-mean tau-PET in Aβ+

**b** Annual group-mean tau-PET change in Aβ+

**c** Model 1: tau-PET change predicted by tau weighted-by distance

**d** Model 2: tau-PET change predicted by tau-weighted functional connectivity

**e** Model 3: tau-PET change predicted by tau- and distance weighted functional connectivity

ADNI

BioFINDER

0 Percentile 100

**Fig. 8 Surface rendering of predicted tau-PET change.** Surface renderings of percentile-transformed group-mean tau-PET at baseline (**a**) and annual tau-PET change (**b**) for Aβ+ subjects of the ADNI (N = 53) and BioFINDER (N = 41) sample. The prediction of tau-PET change from Models 1–3 shown in Fig. 7 is illustrated via surface renderings in (**c**–**e**).

is indeed associated with similar tau accumulation rates. This finding recapitulates in vitro and rodent experiments showing that optogenetic connectivity enhancement of tau-harboring neurons leads to faster tau accumulation in connected regions[15,16]. Similar to these preclinical findings, we show that higher connectivity of fast tau accumulating regions is associated with faster tau accumulation in connected regions. In contrast, higher connectivity of slow tau accumulating regions is associated with slower tau

accumulation in connected regions. This finding echoes our previous cross-sectional findings in AD, where inferior temporal tau hotspots were preferentially connected to other high-tau regions, whereas low-tau regions (e.g. in the motor cortex) were preferentially connected to other low-tau regions[19]. In a similar vein, other studies have shown that grey matter atrophy of inferior temporal regions—i.e. a downstream consequence of accumulating tau pathology—is associated with atrophy in connected regions[29], and that connectivity based models can in general predict progression of brain atrophy in AD[30]. Together, it is thus possible that functional connectivity of fast tau accumulating regions may facilitate the spread of tau to their closely connected neighbors. An alternative explanation is, that network-forming regions show similar susceptibility for developing tau pathology: Specifically, previous studies have shown that similar gene expression is found among functionally connected regions[31,32], where similar gene expression across brain regions is associated with shared susceptibility to develop AD pathology, including amyloid, tau and neurodegeneration[33–35]. Thus, it is possible that the association between functional connectivity and tau accumulation rates is in part determined by shared genetic susceptibility for developing tau pathology. Since both explanations are not exclusive, similar tau accumulation rates may be found preferentially along functionally connected regions due to both connectivity-mediated tau spreading as well as shared genetic susceptibility for developing tau pathology.

Further, we could show that connectivity at baseline is associated with future tau accumulation both on the group- and subject-level. Specifically, we could show that the rate of future tau accumulation of any given target brain region can be modeled by a combination of its current connectivity strength to other tau-seeding brain regions, their respective tau load and the approximate length of the connections. In contrast, taking into account only proximity between brain regions did not (BioFINDER) or hardly (ADNI) explain tau accumulation rates. Thus, our result pattern supports the idea that tau pathology spreads throughout the brain network, where functionally strong and spatially short connections increase the likelihood of tau seeding and spread. In other words, our results support the view that tau spreading is determined primarily by connectivity and not by proximity[13], where tau spreading is assumed to be an active process along connected brain regions rather than passive diffusion[7]. Our findings have important clinical implications: Knowledge of future tau spread can be critical for predicting clinical disease progression, since the level of tau is the strongest predictor of cognitive impairment and cognitive decline in AD[36,37]. Here, blocking tau spread by targeting neural activity related mechanisms of tau spread could be a promising target for attenuating AD progression[21], especially in view of the limited clinical efficacy of anti-amyloid trials[38,39].

Several caveats should be considered, however, when interpreting the results of the current study. First, the AV1451 tau-PET tracer shows considerable unspecific off-target binding in brain regions like the basal ganglia, hippocampus and choroid plexus, which may confound the current joint analyses on tau-PET and functional connectivity[40]. To address this, we preselected an atlas that excludes these typical AV1451 off-target binding regions. Still, it is possible, that unspecific AV1451 binding may influence our results, hence our findings await further replication once second-generation tau-PET tracers with a better off-target binding profile are available[41,42]. Here, it will be of special interest to study the role of the hippocampus in connectivity-related tau-spreading, which is a site of early tau pathology that may be critically involved in tau spread from allo- to the neocortical regions[43,44].

Second, partial-volume-corrected AV1451 tau-PET data were only available in BioFINDER, given the limited availability of concurrent longitudinal MRI scans matching the longitudinal tau-PET acquisitions in ADNI. Thus, all results reported in this manuscript are nonpartial volume-corrected, and usage of a single MRI for coregistration of the PET data in ADNI may introduce asymmetry bias[45]. However, repeating all analyses in BioFINDER with partial-volume-corrected data yielded consistent results with results reported in this manuscript (data not shown). Congruent with this result pattern, previous studies have shown that longitudinal AV1451 tau-PET changes can be detected in ROI based analyses without partial-volume correction, where partial-volume correction simply increases the sensitivity for detecting AV1451 tau-PET changes[27]. Similarly, recent studies using voxel-wise approaches have also reported significant tau accumulation in both aging and AD in nonpartial volume-corrected tau-PET data[46,47]. Together, partial-volume correction may enhance the sensitivity to detect tau-PET changes, but tau-PET changes can also be detected without partial-volume correction. Thus, our findings are unlikely to be driven by partial-volume effects.

Third, the current study focuses solely on functional connectivity, i.e. shared neuronal activity between brain regions, which is to a large degree but not entirely matched by structural connectivity as assessed via diffusion tensor imaging[48]. This mismatch between functional and structural connections may in part be determined by technical limitations of DTI to detect crossing-fibers or short-range cortico-cortical connections[49]. On the other hand, the slow temporal resolution of resting-state fMRI may lead to connectivity between brain regions that exhibit no direct but multi-synaptic connections[50]. Thus, it is important to keep in mind that our current results on functional connectivity and covariance in tau accumulation likely reflect a mixture of direct and indirect connections between brain regions. Here, future studies may combine both DTI and fMRI in order to assess the joint contribution of structural and functional connectivity to tau spreading. Importantly, future methodological advances in MRI-based connectomics may lead to a better matching of functional and structural connectivity, which may—in combination with tau PET—further our understanding on the association between connectivity tau spreading.

In conclusion, the current study demonstrates that connectivity is associated with future tau spread in AD. Our independently validated findings provide strong support for the notion that tau spreads across synaptic connections in an activity-dependent manner, as suggested by preclinical findings[14–16]. Our results may also motivate future studies to investigate tau spreading in other tauopathies such as primary age-related tauopathy, progressive nuclear palsy or corticobasal degeneration where tau pathology may spread via similar mechanisms[51–53]. Since tau is the strongest driver of cognitive decline in AD, potential interventions could target connectivity-related spreading mechanisms[16] or silencing of amyloid-induced neuronal hyperactivity that may promote tau spreading[54,55]. Together, limiting tau spreading across brain connections may be a promising approach to slow AD progression.

## Methods

**ADNI participants**. We included 81 participants from ADNI phase 3 (ClinicalTrials.gov ID: NCT02854033) based on availability of baseline T1-weighted & resting-state fMRI, [18]F-AV45 amyloid-PET and at least two [18]F-AV1451 tau-PET visits. The T1-weighted, resting-state fMRI, AV45 amyloid-PET and the first AV1451 image had to be obtained within the same study visit. Using Freesurfer-derived global AV45 amyloid-PET SUVR scores normalized to the whole cerebellum (provided by the ADNI-PET Core), all subjects were characterized as Aβ+ or Aβ− based on established cut-points (global AV45 SUVR > 1.11)[56]. For the Aβ- group, we included 28 cognitively normal subjects (CN, MMSE > 24, CDR = 0, nondepressed). To cover the spectrum of AD, we included 32 CN and 21 mild cognitively impaired subjects (MCI; MMSE > 24, CDR = 0.5, objective memory-loss on the education adjusted Wechsler Memory Scale II, preserved activities

of daily living) with elevated amyloid deposition (i.e. Aβ+, global AV45 SUVR > 1.11)[57]. Ethical approval was obtained by the ADNI investigators at each participating ADNI site, all participants provided written informed consent.

**BioFINDER participants.** As an independent validation sample, we included 57 participants from the BioFINDER cohort, that were selected based on availability of amyloid-status, longitudinal AV1451 tau-PET and structural MRI data. Amyloid-status of all subjects was determined at the baseline visit via Flutemetamol-PET as described previously[58], applying a pons-normalized global SUVR cut-off of 0.575[59]. Within this sample, the spectrum of AD was covered by 16 CN Aβ+, 7 MCI Aβ+ and 18 AD dementia subjects. As a control sample, we included 16 CN Aβ− subjects. Inclusion and exclusion criteria as well as diagnostic criteria within the BioFINDER cohort have been described previously[60]. All participants signed a written informed consent to participate in the study prior to inclusion in the study. Ethical approval was given by the regional ethics committee at Lund University, Sweden. Imaging procedures were approved by the Radiation protection committee at Skåne University Hospital and by the Swedish Medical Products Agency.

**MRI and PET acquisition and preprocessing in ADNI.** In ADNI, all MRI data were obtained on 3T scanners using unified scanning protocols (parameter details can be found on: https://adni.loni.usc.edu/wp-content/uploads/2017/07/ADNI3-MRI-protocols.pdf). Structural MRI was recorded using a 3D T1-weighted MPRAGE sequence with 1 mm isotropic voxel-size and a TR = 2300 ms. For functional MRI, for each subject a total of 200 resting-state fMRI volumes were recorded using a 3D EPI sequence in 3.4 mm isotropic voxel resolution with a TR/TE/flip angle = 3000/30/90°.

Tau-PET was assessed in 6 × 5 min blocks 75 min after intravenous bolus injection of [18]F-radiolabeled AV1451. All tau-PET images were intensity normalized using the inferior cerebellar grey as a reference region following a previously described protocol[26]. Usage of an alternative reference region (i.e. eroded white-matter[46]), yielded consistent results with the analyses presented in this manuscript. Nonlinear spatial normalization parameters were estimated based on structural skull-stripped baseline T1-weighted images using Advanced Normalization Tools (ANTs), to normalize all images to Montreal Neurological Institute (MNI) standard space. The two AV1451 images were then coregistered to the native space baseline T1-MRI image and subsequently normalized to MNI space by applying the ANTs-derived spatial normalization parameters.

For ADNI resting-state fMRI images, we first applied motion correction (i.e. realignment), regressed out the mean signal from the white matter, cerebrospinal fluid as well as the six motion parameters that were estimated during realignment (i.e. three translations and three rotations). Next we applied detrending, band-pass filtering (0.01–0.08 Hz) and despiking. To further eliminate motion artifacts, we performed scrubbing, i.e. removal of high-motion frames as defined by exceeding 0.5 mm framewise displacement. Specifically, high-motion volumes together with one preceding and two subsequent volumes were replaced with zero-padded volumes to eliminate high-motion volumes but keep the number of volumes consistent across subjects. Lastly, the preprocessed resting-state fMRI images were spatially normalized to MNI space by (1) coregistration to the baseline T1-weighted images followed by applying the ANTs-derived nonlinear transformation parameters.

Note that we did not perform global signal regression due to some controversies about potential bias introduced by this preprocessing step[61,62]. However, when reanalyzing the data with global signal regression, all results presented in this manuscript remained virtually the same.

**MRI and PET acquisition and preprocessing in BioFINDER.** In BioFINDER, 1 mm isotropic T1-weighted MPRAGE (TR/TE = 1900/2.64 ms) and Fluid-attenuated inversion recovery (FLAIR; 0.7 × 0.7 × 5 mm³, 23 slices, TR/TE = 9000/81 ms) MRI images were acquired for all participants on a 3T Siemens Skyra scanner (Siemens Medical Solutions, Erlangen, Germany). Tau-PET imaging was conducted 80–100 min after bolus injection of [18]F-Flortaucipir on a GE Discovery 690 PET scanner (General Electric Medical Systems, Milwaukee, WI, USA). Radiosynthesis and radiochemical purity for [18]F-AV1451 within the BioFINDER study have previously been described in detail[63]. The image data were processed by the BioFINDER imaging core using a pipeline developed at Lund University that was described previously[64]. In brief the MRIs were skull stripped using the combined MPRAGE and FLAIR data, segmented into grey and white matter and normalized to MNI space. PET images were attenuation corrected, motion corrected, summed and coregistered to the MRIs. In line with the ADNI data, standardized uptake value ratio (SUVR) data were calculated using an inferior cerebellar grey matter as reference region. Usage of an alternative reference region (i.e. eroded white matter) yielded consistent results with the analyses reported in the manuscript. Both nonpartial-volume-corrected data and data corrected for partial-volume using the geometrical transfer matrix method[65] were calculated. Usage of partial-volume-corrected data yielded consistent results with the results obtained on nonpartial volume-corrected data that are reported in the current manuscript.

To determine a functional connectivity template for the BioFINDER sample, we downloaded spatially normalized (i.e. to MNI space) minimally preprocessed resting-state fMRI images from 500 subjects of the human-connectome project (HCP). Congruent with the preprocessing approach in ADNI, we further applied detrending, band-pass filtering (0.01–0.08 Hz), despiking and motion scrubbing to the HCP resting-state data.

**Functional connectivity and covariance in tau change.** FC and covariance in tau change were estimated in an ROI based manner, using 400 ROIs from the Schaefer fMRI atlas (see Fig. 2a), that is based on a data driven fMRI brain parcellation[23]. The 400 ROIs can be grouped within seven large-scale functional networks that match well with previously described parcellations[66]. This atlas is especially well-suited for joint analyses of AV1451 tau-PET and fMRI, since the 400 ROIs cover only the neocortex and exclude typical AV1451 off-target binding regions like the hippocampus or basal ganglia[23,67], which may otherwise confound the results. Prior to all analyses, the Schaefer fMRI atlas was masked with a grey matter mask[68]. To later address whether the distance between ROIs can explain any associations between FC and tau, we further computed the Euclidean distance between ROIs, defined as the geometric distance between the center of mass of each ROI.

To ensure that the results reported in this manuscript were not driven by the nature of the 400 ROI brain parcellation all analyses were using the 200 ROI variant of the same atlas[23]. Using the 200-ROI parcellation, the result pattern that was consistent with all results reported in this manuscript. The results of this confirmatory analysis can be found in Supplementary Figs. 1–5.

**Functional connectivity assessment.** In ADNI, FC was estimated for each subject based on the preprocessed fMRI data from which the mean fMRI time-course was extracted for each of the 400 ROIs by averaging the signal across ROI-specific per volume. Using these 400 ROI-specific timecourses, we assessed functional connectivity as Fisher-z transformed Pearson-moment correlations between all possible ROI pairs. Autocorrelations were set to zero, positive and negative connectivity values were retained. For each group (i.e. CN Aβ−, Aβ+), we then computed group-average FC matrices across subjects. To determine a functional connectivity template for the BioFINDER sample, we used the above described methods to determine a group-average functional connectivity matrix from the preprocessed fMRI data of the 500 HCP subjects. For the group-average ADNI and HCP functional connectivity matrices, we further determined 200 null-models of functional connectivity respectively, by shuffling the functional connectivity matrices while preserving the overall degree- and weight-distribution, using the null_model_und_sign.m function of the brain connectivity toolbox (https://sites.google.com/site/bctnet/). These shuffled connectivity matrices were later used to determine whether associations between functional connectivity and longitudinal tau-PET changes are driven by the topological structure of functional connectivity and not by the mere statistical properties of functional connectivity in general. Exploratorily restricting all analyses to positive connectivity values only yielded congruent results with the analyses presented in the manuscript.

**Assessing covariance in tau change.** For both the ADNI and BioFINDER sample, we assessed the correlation between the levels of tau-PET change in a given region X and another region Y (see Fig. 3a for an analysis flow-chart). The analysis pipeline was adopted in a modified way from our previous study where we used cross-sectional tau-PET data to determine tau covariance. In a similar vein, others have used this approach to determine FDG-PET (i.e. metabolic connectivity) or grey matter covariance (i.e. structural connectivity) across brain regions[69,70]. In a first step, we computed the mean annual tau-PET change within each of the 400 ROIs for each subject within the ADNI and BioFINDER sample. This was done by computing for each ROI the baseline vs. follow-up tau-PET difference divided by the time (in years) between the two AV1451 tau-PET scans, yielding a 400-element vector of annual tau-PET change per subject. Using these 400-element tau-PET change vectors, we then assessed within Aβ+ subjects of the ADNI or BioFINDER across subjects the pairwise ROI-to-ROI Spearman correlation of tau-PET change. We specifically used Spearman correlation to avoid that the estimation of ROI-to-ROI covariance in tau change estimation was driven by strong changes in single subjects or ROIs. This analysis resulted in a single 400 × 400 sized covariance in tau change matrix for Aβ+ subjects of both the ADNI and BioFINDER sample (Fig. 3b). Again, autocorrelations were set to zero and all correlations were Fisher-z transformed, equivalent to the FC matrices. For each sample, an additional covariance in tau change matrix controlled for age, sex, ApoE4 and education was obtained using linear regression for each ROI pair.

**Statistics.** Group differences in baseline demographics were assessed using ANOVAs for continuous and Chi-squared tests for nominal data. For descriptive purposes, significant annual tau-PET changes were assessed using ROI-wise one sample t-tests, applying an uncorrected alpha threshold of 0.005. Group differences in the 400 × 400 functional connectivity matrices between CN Aβ− and Aβ+ groups in ADNI were assessed using element-wise ANCOVAs controlling for age, sex, education and diagnosis, applying a Bonferroni-corrected α-threshold of 0.05.

To test our main hypothesis (i.e. association between functional connectivity and covariance in tau change), we applied linear regression, with the vectorized

group-average functional connectivity matrix (i.e using ADNI functional connectivity data for the ADNI sample and HCP functional connectivity data for BioFINDER) as a predictor of the vectorized covariance in tau change matrix. For exploratory reasons, we also assessed the association between covariance in tau-PET change and functional connectivity separately for each of the seven canonical brain networks. The association between whole-brain functional connectivity and covariance in tau-PET change was further determined for ADNI and BioFINDER using the 200 respective shuffled ADNI and HCP functional connectivity null-models, to obtain a null-distribution of the β-values that was used to compare the true β−value using an exact test. In ADNI, we further assessed the robustness of the association between functional connectivity and covariance in tau-PET change via bootstrapping. Specifically, we drew 1000 random samples with replacement from the entire group of 53 Aβ+ subjects and assessed for each sample the group-mean functional connectivity, covariance in tau-PET change, as well as the association between them. By saving the 1000 bootstrapping derived β-values we obtained the 95% CI and tested whether the β-value distribution deviated from zero. Note that this bootstrapping approach was exclusively conducted in ADNI, since it required availability of both subject-specific functional connectivity and tau-PET data. The above described whole-brain analyses were further repeated while additionally controlling the regression model for Euclidean distance between each ROI pair, to assess whether associations between functional connectivity and covariance in tau change were independent of distance. Also, we repeated the whole-brain analyses using covariate controlled (i.e. age, education, sex, ApoE4-status, diagnosis and MMSE score) covariance in tau-PET matrices, to ensure that the association between functional connectivity and covariance in tau-PET change was not driven by these covariates.

In a next step, we tested whether the level of tau-PET change in a given seed ROI is associated with the tau-PET changes in closely connected regions. The rationale is that if tau spreads as a function of functional connectivity, then ROIs with similar tau changes should be connected. To test this, we rank-ordered all ROIs according to their level of tau-PET change. Using linear regression, we tested for each rank-ordered ROI (seed), the group-average functional connectivity to the remaining ROIs (target) as a predictor of the group-average level of tau-PET change in the target ROIs (Fig. 5a). Again, we performed the same analyses using the 200 shuffled connectomes, to compare the true β-value with a β-value null-distribution using an exact test. In ADNI, we further determined the robustness of this analysis by repeating the entire procedure using the above described bootstrapping procedure with 1000 randomly drawn samples from the overall pool of 53 Aβ+ subjects, based on which group-average tau-PET change and functional connectivity were iteratively determined.

Lastly, we tested whether future tau change can be modeled by functional connectivity and tau load at baseline, using three approaches. As a negative control, we tested whether tau spread is a function of baseline tau and the Euclidean distance between ROIs. Specifically, we determined the mean tau-weighted Euclidean distance between a given tau-receiving target ROI and all other tau-seeding 399 seed ROIs, after multiplying each of the 399 distance values by the respective seed ROIs baseline tau-PET level (Fig. 7a, Model 1). For our second approach, we tested whether tau spread can be modeled by combining tau at baseline and functional connectivity. Specifically, we computed the mean functional connectivity between a given tau-receiving target ROI and all other tau-sending 399 seed ROIs, after multiplying each of the 399 functional connectivity values by the respective seed ROIs baseline tau level (Fig. 7a, Model 2). Third, we tested whether adding Euclidean distance as an additional multiplication factor in the above listed model could further improve the association strength with future tau spread (Fig. 7a, Model 3).

Within the ADNI and BioFINDER samples, each approach yielded a 400-element vector that was tested as a predictor of annual tau-PET changes in the corresponding 400 ROIs via linear regression. Again, we conducted the same analyses using the shuffled connectomes to compare the true β-value with a β-value null-distribution using an exact test. In ADNI, we further performed bootstrapping using 1000 samples based on which group-average functional connectivity and tau-PET change were iteratively determined as described above. Within the ADNI, the bootstrapped β-value distributions for each of the three approaches were then compared using an ANOVA, to determine which approach yielded the most accurate association with longitudinal tau changes.

Next, we assessed prediction model performance on the subject-level. For ADNI, we used subject-level tau-PET and functional connectivity data, and for BioFINDER we used subject-level tau-PET and group-level HCP functional connectivity data. Prediction model performance (i.e. β-values reflecting the association between predicted and actual tau-PET changes) was compared across models using an ANCOVA. Here, we also tested whether prediction performance (i.e. model-derived β-values) was associated with age (using linear regression) or with sex and ApoE4 status (using ANOVAs).

**Reporting summary**. Further information on research design is available in the Nature Research Reporting Summary linked to this article.

## Data availability

The data that used in this study were obtained from the Alzheimer's disease Neuroimaging Initiative (ADNI) and are available from the ADNI database (adni.loni.

usc.edu) upon registration and compliance with the data usage agreement. Data from the BioFINDER sample are available from the authors upon request. Resting-state data of the HCP cohort are freely available online (https://db.humanconnectome.org). Source data underlying Figs. 4, 6, 7 is available as a Source Data file.

## Code availability

An example version of the R-Markdown code used for the main analysis can be found (together with simulated data) in the supplementary.

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

## Acknowledgements

Parts of the data used in preparation of this manuscript were obtained from the ADNI database (adni.loni.usc.edu). As such, the investigators within the ADNI study contributed to the design and implementation of ADNI and/or provided data but did not participate in analysis or writing of this paper. A complete listing of ADNI investigators can be found at the end of the manuscript. The study was funded by grants from the Alzheimer Forschung Initiative (AFI, Grant 15035 to M.E.), European Commission (Grant 334259 to M.E.), LMU excellence (AOST 865297 to M.E.) and Förderung Forschung Lehre (1032 to N.F.). ADNI data collection and sharing for this project was funded by the ADNI (National Institutes of Health Grant U01 AG024904) and DOD ADNI (Department of Defense award number W81XWH-12-2-0012). ADNI is funded by the National Institute on Aging, the National Institute of Biomedical Imaging, and Bioengineering, and through contributions from the following: AbbVie, Alzheimer's Association; Alzheimer's Drug Discovery Foundation; Araclon Biotech; BioClinica, Inc.; Biogen; Bristol-Myers Squibb Company; CereSpir, Inc.; Cogstate; Eisai Inc.; Elan Phar- maceuticals, Inc.; Eli Lilly and Company; EuroImmun; F. Hoffmann-La Roche Ltd and its affiliated company Genentech, Inc.; Fujirebio; GE Healthcare; IXICO Ltd.; Janssen Alzheimer Immunotherapy Research & Development, LLC.; Johnson & Johnson Phar- maceutical Research & Development LLC.; Lumosity; Lundbeck; Merck & Co., Inc.; Meso Scale Diagnostics, LLC.; NeuroRx Research; Neurotrack Technologies; Novartis Pharmaceuticals Corporation; Pfizer Inc.; Piramal Imaging; Servier; Takeda Pharma- ceutical Company; and Transition Therapeutics. The Canadian Institutes of Health Research is providing funds to support ADNI clinical sites in Canada. Private sector contributions are facilitated by the Foundation for the National Institutes of Health (www.fnih.org).

## Author contributions

N.F.: study concept and design, data processing, statistical analysis, interpretation of the results, writing the manuscript. A.R.: data processing, critical revision of the manuscript. J.N.: data processing, critical revision of the manuscript. R.S.: critical revision of the manuscript. O.S.: critical revision of the manuscript. R.O.: critical revision of the

manuscript. O.H.: critical revision of the manuscript. M.E.: study concept and design, interpretation of the results, writing the manuscript. ADNI provided all data used for this study.

## Competing interests
The authors declare no competing interests.

## Additional information

## Alzheimer's Disease Neuroimaging Initiative (ADNI)

Michael Weiner[6], Paul Aisen[7], Ronald Petersen[8], Clifford R. Jack Jr.[8], William Jagust[9], John Q. Trojanowki[10], Arthur W. Toga[11], Laurel Beckett[12], Robert C. Green[13], Andrew J. Saykin[14], John Morris[15], Leslie M. Shaw[16], Enchi Liu[17], Tom Montine[18], Ronald G. Thomas[7], Michael Donohue[7], Sarah Walter[7], Devon Gessert[7], Tamie Sather[7], Gus Jiminez[7], Danielle Harvey[12], Michael Donohue[7], Matthew Bernstein[8], Nick Fox[19], Paul Thompson[20], Norbert Schuff[21], Charles DeCArli[12], Bret Borowski[22], Jeff Gunter[22], Matt Senjem[22], Prashanthi Vemuri[22], David Jones[22], Kejal Kantarci[22], Chad Ward[22], Robert A. Koeppe[23], Norm Foster[24], Eric M. Reiman[25], Kewei Chen[25], Chet Mathis[26], Susan Landau[9], Nigel J. Cairns[15], Erin Householder[15], Lisa Taylor Reinwald[15], Virginia Lee[27], Magdalena Korecka[27], Michal Figurski[27], Karen Crawford[11], Scott Neu[11], Tatiana M. Foroud[14], Steven Potkin[28], Li Shen[14], Faber Kelley[14], Sungeun Kim[14], Kwangsik Nho[14], Zaven Kachaturian[29], Richard Frank[30], Peter J. Snyder[31], Susan Molchan[32], Jeffrey Kaye[33], Joseph Quinn[33], Betty Lind[33], Raina Carter[33], Sara Dolen[33], Lon S. Schneider[34], Sonia Pawluczyk[34], Mauricio Beccera[34], Liberty Teodoro[34], Bryan M. Spann[34], James Brewer[35], Helen Vanderswag[35], Adam Fleisher[35], Judith L. Heidebrink[23], Joanne L. Lord[23], Ronald Petersen[8], Sara S. Mason[8], Colleen S. Albers[8], David Knopman[8], Kris Johnson[8], Rachelle S. Doody[36], Javier Villanueva Meyer[36], Munir Chowdhury[36], Susan Rountree[36], Mimi Dang[36], Yaakov Stern[37], Lawrence S. Honig[37], Karen L. Bell[37], Beau Ances[38], John C. Morris[38], Maria Carroll[38], Sue Leon[38], Erin Householder[38], Mark A. Mintun[38], Stacy Schneider[38], Angela OliverNG[39], Randall Griffith[39], David Clark[39], David Geldmacher[39], John Brockington[39], Erik Roberson[39], Hillel Grossman[40], Effie Mitsis[40], Leyla deToledo-Morrell[41], Raj C. Shah[41], Ranjan Duara[42], Daniel Varon[42], Maria T. Greig[42], Peggy Roberts[42], Marilyn Albert[43], Chiadi Onyike[43], Daniel D'Agostino II[43], Stephanie Kielb[43], James E. Galvin[44], Dana M. Pogorelec[44], Brittany Cerbone[44], Christina A. Michel[44], Henry Rusinek[44], Mony J de Leon[44], Lidia Glodzik[44], Susan De Santi[44], P. Murali Doraiswamy[45], Jeffrey R. Petrella[45], Terence Z. Wong[45], Steven E. Arnold[16], Jason H. Karlawish[16], David Wolk[16], Charles D. Smith[46], Greg Jicha[46], Peter Hardy[46], Partha Sinha[46], Elizabeth Oates[46], Gary Conrad[46], Oscar L. Lopez[26], MaryAnn Oakley[26], Donna M. Simpson[26], Anton P. Porsteinsson[47], Bonnie S. Goldstein[47], Kim Martin[47], Kelly M. Makino[47], M. Saleem Ismail[47], Connie Brand[47], Ruth A. Mulnard[48], Gaby Thai[48], Catherine Mc Adams Ortiz[48], Kyle Womack[49], Dana Mathews[49], Mary Quiceno[49], Ramon Diaz Arrastia[49], Richard King[49], Myron Weiner[49], Kristen Martin Cook[49], Michael DeVous[49], Allan I. Levey[50], James J. Lah[50], Janet S. Cellar[50], Jeffrey M. Burns[51],

Heather S. Anderson[51], Russell H. Swerdlow[51], Liana Apostolova[52], Kathleen Tingus[52], Ellen Woo[52], Daniel H.S. Silverman[52], Po H. Lu[52], George Bartzokis[52], Neill R Graff Radford[53], Francine ParfittH[53], Tracy Kendall[53], Heather Johnson[53], Martin R. Farlow[14], Ann Marie Hake[14], Brandy R. Matthews[14], Scott Herring[14], Cynthia Hunt[14], Christopher H. van Dyck[54], Richard E. Carson[54], Martha G. MacAvoy[54], Howard Chertkow[55], Howard Bergman[55], Chris Hosein[55], Sandra Black[56], Bojana Stefanovic[56], Curtis Caldwell[56], Ging Yuek Robin Hsiung[57], Howard Feldman[57], Benita Mudge[57], Michele Assaly Past[57], Andrew Kertesz[58], John Rogers[58], Dick Trost[58], Charles Bernick[59], Donna Munic[59], Diana Kerwin[60], Marek Marsel Mesulam[60], Kristine Lipowski[60], Chuang Kuo Wu[60], Nancy Johnson[60], Carl Sadowsky[61], Walter Martinez[61], Teresa Villena[61], Raymond Scott Turner[62], Kathleen Johnson[62], Brigid Reynolds[62], Reisa A. Sperling[63], Keith A. Johnson[63], Gad Marshall[63], Meghan Frey[63], Jerome Yesavage[64], Joy L. Taylor[64], Barton Lane[64], Allyson Rosen[64], Jared Tinklenberg[64], Marwan N. Sabbagh[65], Christine M. Belden[65], Sandra A. Jacobson[65], Sherye A. Sirrel[65], Neil Kowall[66], Ronald Killiany[66], Andrew E. Budson[66], Alexander Norbash[66], Patricia Lynn Johnson[66], Thomas O. Obisesan[67], Saba Wolday[67], Joanne Allard[67], Alan Lerner[68], Paula Ogrocki[68], Leon Hudson[68], Evan Fletcher[69], Owen Carmichael[69], John Olichney[69], Charles DeCarli[69], Smita Kittur[70], Michael Borrie[71], T Y Lee[71], Rob Bartha[71], Sterling Johnson[72], Sanjay Asthana[72], Cynthia M. Carlsson[72], Steven G. Potkin[73], Adrian Preda[73], Dana Nguyen[73], Pierre Tariot[25], Adam Fleisher[25], Stephanie Reeder[25], Vernice Bates[74], Horacio Capote[74], Michelle Rainka[74], Douglas W. Scharre[75], Maria Kataki[75], Anahita Adeli[75], Earl A. Zimmerman[76], Dzintra Celmins[76], Alice D. Brown[76], Godfrey D. Pearlson[77], Karen Blank[77], Karen Anderson[77], Robert B. Santulli[78], Tamar J. Kitzmiller[78], Eben S. Schwartz[78], Kaycee M. SinkS[79], Jeff D. Williamson[79], Pradeep Garg[79], Franklin Watkins[79], Brian R. Ott[80], Henry Querfurth[80], Geoffrey Tremont[80], Stephen Salloway[81], Paul Malloy[81], Stephen Correia[81], Howard J. Rosen[6], Bruce L. Miller[6], Jacobo Mintzer[82], Kenneth Spicer[82], David Bachman[82], Elizabether Finger[83], Stephen Pasternak[83], Irina Rachinsky[83], John Rogers[83], Andrew Kertesz[83], Dick Drost[83], Nunzio Pomara[84], Raymundo Hernando[84], Antero Sarrael[84], Susan K. Schultz[85], Laura L. Boles Ponto[85], Hyungsub Shim[85], Karen Elizabeth Smith[85], Norman Relkin[86], Gloria Chaing[86], Lisa Raudin[86], Amanda Smith[87], Kristin Fargher[87] & Balebail Ashok Raj[87]

[6]UC San Francisco, San Francisco, CA, USA. [7]UC San Diego, San Diego, CA, USA. [8]Mayo Clinic, Rochester, NY, USA. [9]UC Berkeley, Berkeley, CA, USA. [10]U Pennsylvania, Pennsylvania, CA, USA. [11]USC, Los Angeles, CA, USA. [12]UC Davis, Davis, CA, USA. [13]Brigham and Women's Hospital, Harvard Medical School, Boston, MA, USA. [14]Indiana University, Bloomington, IND, USA. [15]Washington University St. Louis, St. Louis, MO, USA. [16]University of Pennsylvania, Philadelphia, PA, USA. [17]Janssen Alzheimer Immunotherapy, South San Francisco, CA, USA. [18]University of Washington, Seattle, WA, USA. [19]University of London, London, UK. [20]USC School of Medicine, Los Angeles, CA, USA. [21]UCSF MRI, San Francisco, CA, USA. [22]Mayo Clinic, Rochester, NY, USA. [23]University of Michigan, Ann Arbor, MI, USA. [24]University of Utah, Salt Lake City, UT, USA. [25]Banner Alzheimer's Institute, Phoenix, AZ, USA. [26]University of Pittsburgh, Pittsburgh, PA, USA. [27]UPenn School of Medicine, Philadelphia, PA, USA. [28]UC Irvine, Newport Beach, CA, USA. [29]Khachaturian, Radebaugh & Associates, Inc and Alzheimer's Association's Ronald and Nancy Reagan's Research Institute, Chicago, IL, USA. [30]General Electric, Boston, MA, USA. [31]Brown University, Providence, RI, USA. [32]National Institute on Aging/National Institutes of Health, Bethesda, MD, USA. [33]Oregon Health and Science University, Portland, OR, USA. [34]University of Southern California, Los Angeles, CA, USA. [35]University of California San Diego, San Diego, CA, USA. [36]Baylor College of Medicine, Houston, TX, USA. [37]Columbia University Medical Center, New York, NY, USA. [38]Washington University, St. Louis, MO, USA. [39]University of Alabama Birmingham, Birmingham, MO, USA. [40]Mount Sinai School of Medicine, New York, NY, USA. [41]Rush University Medical Center, Chicago, IL, USA. [42]Wien Center, Vienna, Austria. [43]Johns Hopkins University, Baltimore, MD, USA. [44]New York University, New York, NY, USA. [45]Duke University Medical Center, Durham, NC, USA. [46]University of Kentucky, city of Lexington, NC, USA. [47]University of Rochester Medical Center, Rochester, NY, USA. [48]University of California, Irvine, CA, USA. [49]University of Texas Southwestern Medical School, Dallas, TX, USA. [50]Emory University, Atlanta, GA, USA. [51]University of Kansas, Medical Center, Lawrence, KS, USA. [52]University of California, Los Angeles, CA, USA. [53]Mayo Clinic, Jacksonville, FL, USA. [54]Yale University School of Medicine, New Haven, CT, USA. [55]McGill Univ., Montreal Jewish General Hospital, Montreal, WI, USA. [56]Sunnybrook Health Sciences, Toronto, ON, Canada. [57]U.B.C. Clinic for AD & Related Disorders, British Columbia, BC, Canada. [58]Cognitive Neurology St. Joseph's, Toronto, ON, Canada. [59]Cleveland Clinic Lou Ruvo Center for Brain Health, Las Vegas, NV, USA. [60]Northwestern University, Evanston, IL, USA. [61]Premiere Research Inst Palm Beach Neurology, West Palm Beach, FL, USA. [62]Georgetown University Medical Center, Washington, DC, USA. [63]Brigham and Women's Hospital, Boston, MA, USA. [64]Stanford University, Santa Clara County, CA, USA. [65]Banner Sun Health Research Institute, Sun City, AZ, USA. [66]Boston University, Boston, MA, USA. [67]Howard University, Washington, DC, USA. [68]Case Western Reserve University, Cleveland, OH, USA. [69]University of California, Davis Sacramento, CA, USA. [70]Neurological Care of CNY, New York, NY, USA. [71]Parkwood Hospital, Parkwood, CA, USA. [72]University of Wisconsin, Madison, WI, USA. [73]University of California, Irvine BIC, Irvine, CA, USA. [74]Dent Neurologic Institute, Amherst, MA, USA. [75]Ohio State University, Columbus, OH, USA. [76]Albany Medical College, Albany, NY, USA.

