## [Peer Review File · Nature Communications]

Reviewers' comments:

Reviewer #1 (Remarks to the Author):

The paper by Franzmeier and colleagues shows that non-demented individuals with amyloid pathology exhibit faster tau accumulation in temporo-parietal regions than controls, which has been shown many times before. However, the authors revealed that functionally connected regions show correlated tau accumulation rates. Further, they found that subsequent tau accumulation in pre-dementia AD could be predicted by a combination of baseline tau levels, functional connectivity and distance between brain regions. The latter findings are of great interest to the field. The paper is generally well written, but there are some major concerns:

- 1) The study included very few individuals (15 Ab- CU and 30 Ab+ CU/MCI) and the follow-up time is very short (appr. 1.4 y). The authors should consequently include an independent validation cohort, which have at least 2.0 y of follow-up. Further, it is unclear how the authors selected the 15+30 cases for their study, considering that there is a much larger number available in ADNI.
- 2) The investigated population does not include any cases with AD dementia, which is also a severe limitation, because it is vital to understand whether the spread of tau follows the same principles also during the dementia stages of the disease.
- 3) The population of Ab- cases should be larger, and the authors should investigate whether the regional spread of tau is associated with functional connectivity, also in cases without Ab pathology (PART).
- 4) When the authors include an independent validation cohort, the data could also be pooled to study whether the prediction models differ in 1) younger vs older individuals, 2) females vs males and 3) APOE4 carriers vs non-carriers.
- 5) The authors should replicate their findings using PVEc data. Further, it is unclear to me whether baseline and follow-up PET scans were always co-registered to the MRI scans closest in time to each PET scan (i.e. the baseline and follow-up MRIs that were closest to FTP PET in time).
- 6) It is very unfortunate that the authors do not include the hippocampus in their analyses, considering the central role of this structure for both brain connectivity and tau-pathology. When using PVEc data, it should not be a problem to include FTP data from hippocampus.
- 7) It is unclear which reference region the authors used for FTP PET. I strongly recommend the authors to show whether they get similar results using either 1) inferior cerebellum, or 2) eroded white matter (PMID: 30597624) as the reference region.
- 8) In Figs 1A-B, I am unsure how the authors established a cutoff of 1.2 SUVR when visualizing the FTP uptake. I would recommend 1.3-2.5 SUVR, which is representing a more correct FTP signal less affected by noise.
- 9) In Figs 1D-E, I think the authors should present the voxels with significant change in FTP uptake compared to baseline (i.e. T-values), and not just group-average longitudinal FTP-PET changes. Panel F (not panel D as written in the legend) show the voxels exhibiting significantly different FTP PET slopes between Ab- CU and Ab+ CU/MCI, and the significant clusters are quite small and scattered indicating that the study sample is small.
- 10) The association between "FC" and "covariance in tau change" is quite weak, as shown in Fig 4A. Could the authors investigate whether other factors like age, sex, APOE, education level is affecting this association.
- 11) Further, I am unsure whether the FC in Ab- CU (Fig 2B left) is used for the analyses (in Fig 4A) or the FC in Ab+ CU/MCI cases. To me it would make more sense to use the FC established in individuals without Tau pathology (i.e. Ab- CU) to avoid risk of circularity (i.e. that Tau pathology affects FC, which is then associated with change in FTP PET uptake). If the authors used the FC from Ab- CU for this comparison, they could probably use a much larger sample of Ab- CU cases from ADNI to establish regional FC. Similarly, a structural connectome could be established using DTI in ADNI, which would substantially strengthen the study.
- 12) Both models building on either Euclidean distance or FC were significantly associated with change in FTP PET. Could the authors test another model which they do not believe in to make sure that not all models become positively associated with spread of tau. E.g. they could study

whether the covariance in FTP PET change is associated with either i) the covariance in cortical thickness or ii) blood flow (ASL).

13) Can the authors build a model based on their results predicting the spread through cortex of tau given different "Tau starting regions", e.g. entorhinal cortex or hippocampus? This could perhaps be nicely visualized, which would make it much easier for the reader to compare and judge the relevance of models 1, 2 and 3.

Reviewer #2 (Remarks to the Author):

From the idea that neural connectivity is related to tau accumulation and propagation, Franzmeier et al. have shown that functional connectivity may predict tau propagation in patients with Alzheimer's disease. In a cohort of 15 healthy and 30 pre-dementia Ab-positive subjects, they used baseline and longitudinal tau-PETs to follow tau accumulation, and baseline resting-state fMRI to observe tau values along functionally connected networks. Firstly, authors proved that tau levels at baseline and in longitudinal follow-up increase among Ab-positive individuals. Moreover, among the Ab-positive group, they found a significant spatial relationship between functional connectivity and tau-signal covariance, i.e. higher tau accumulation was seen in more functionally connected areas (according to baseline fMRI). Moreover, Franzmeier et al. showed that this association has predictive value, depending on tau-PET annual change. Particularly, high connectivity relates to high tau-PET change over time in specific target areas only when tau-PET change is high in seed areas, while high connectivity shows the opposite association when tau-PET change is low in seed areas. After confirming that tau spreading is not independent of functional connectivity (negative control), the group created a predictor model for tau propagation based on both functional connectivity and Euclidean distance, as they proved to be better predictors than distance or connectivity separately. Therefore, authors show that tau spreading seems to be an active process dependent on connectivity and connectivity distance.

Overall, the design and execution of this study are sound and findings are very appealing. I have a few suggestions mostly regarding the methods' section of the manuscript.

1. My main concern comes from the spatial regression analysis used in the study. I think this approach is very informative for the purposes of this investigation but authors should explain in more detail how the bootstrapping approach was accomplished. In the present version, authors just mention that the bootstrapping strategy was "iteratively determined via sampling", with no information about how this was done. If I understood well the underlying idea, authors have performed a bootstrapping approach to provide spatial regressions with the actual p-values. As this type of regression (spatial correlations) uses non-independent data (e.g. ROIs are spatially related and therefore non independent from each other), a bootstrapping strategy is necessary to build null hypothesis distributions and avoid the violation of independency between dots in the regressions. However, authors provided little detail about the bootstrapping strategy, which limits the evaluation of this critical step. Moreover, I think authors should delete the initial p-value provided in the (e. g. page 7), as these ones are not p-values derived from null hypothesis distributions.

2. It is bit confusing the explanation (or implementation?) of the neuroimaging scrubbing approach to address the micro-movement. In conventional approaches, researchers either keep the interpolated volumes or remove volumes with high movement and equalize the final number of volumes among all individuals. As far as I can read in this section, authors have used different final number of volumes in each individual. Please clarify, as this may introduce a misbalance in the degrees of freedom across the sample.

3. Please clarify in the text (methods' section) how negative correlations were handling in the processing of the functional connectivity data.

4. Please rephrase the intro sentence that includes: "...has been never tested...", as this is not itself a criteria of novelty.

5. Please replace the term "infect" in the intro.

6. Finally, it is unclear to this reviewer how authors' interpret their "distance" results in the discussion. Is it that more Euclidean distance is positively associated to tau spreading? or less distance is positively associated to tau spreading?

Reviewer #3 (Remarks to the Author):

Franzemeier and colleagues analyze longitudinal tau-PET data in a small sample of non-demented individuals from the ADNI3 dataset, and compare whole-brain regional covariance in tau accumulation with whole-brain regional functional connectivity. Building on their recent publication using cross-sectional data (Franzmeier et al., 2019 Brain), the authors show that regional covariance patterns in longitudinal tau-PET accumulation also correlates with regional functional connectivity, adding to increasing evidence that tau pathology may be transmitted through synaptic connections in human Alzheimer's disease.

This is a compact and focused study with a good introduction, concerning a timely topic of great importance to the field of Alzheimer's disease and dementia. However, it is not clear to me that the results presented here provide a substantial advance over the results of Franzmeier et al., 2019 Brain. While finding similar results using longitudinal data is encouraging, I have additional concerns regarding certain methodological choices and incomplete reporting of crucial methodological details. Please find detailed list of my concerns below:

1) My biggest conceptual concern is that the results presented here do not differ much from Franzmeier et al., 2019 Brain. While longitudinal data is introduced, the sample size is smaller and far less representative of the Alzheimer's disease spectrum, representing only the earliest manifestation of the disease. The cross-sectional and longitudinal patterns from Fig. 1 are largely consistent with Harrison et al., 2019; Jack et al., 2018, Pontecorvo et al. 2019, and the main results are highly consistent with Franzmeier et al. 2019. While a spreading model is introduced in Figure 6, this model is well behind the state of the art of spreading models, such as those presented in Raj et al. 2012 Neuron; Iturria-Medina et al. 2014 PLoS Comp. Biol; Misić et al., 2015 Neuron; Zheng...Dagher 2018 BioRxiv; Olivier...Fleming 2019 BioRxiv; etc. All in all, it is not clear that the addition of longitudinal data adds much new information about the relationship between tau spreading and connectivity in humans.

METHODS

2) The authors do not provide sufficient detail in their PET preprocessing methods to properly evaluate their longitudinal pipeline. The authors for example do not mention which reference region they use, a topic of controversy for longitudinal tau-PET (see e.g. Harrison et al. 2019). How were the reference regions derived (native space or template space)? Does each subject only have one structural MRI? Are T1s segmented or masked before spatial normalization? Is the tau-PET data smoothed? Please provide full details of the methodological pipeline

3) The small sample size remains an issue, but may be partially abated by not restricting the sample to subjects with rsfMRI scans. These functional scans are hardly used at the individual level anyway, and can be substituted with a template connectome based on other ADNI subjects and/or other datasets. Certainly more longitudinal ADNI tau scans are currently available than are included in the present analysis.

4) The authors mention using a Spearman's correlation to compute regional covariance in tau accumulation, an appropriate choice given the sample size and data. However, these values are then Fisher's r-to-z transformed, which will pull out the tails of the r distribution, essentially achieving the opposite goal of the Spearman's correlation by upweighting more extreme values. How do the authors reconcile these contradicting approaches?

5) The fMRI motion correction seems a bit too minimalist, particularly given the clinical cohort and the lack of global signal regression. The authors do not, for example, regress motion parameters out of the time series. This may seem like a small point, but with two low-resolution/smooth measurements, there is a high risk that autocorrelation (caused by e.g. motion) may be driving correlations.

6) Why are different brain masks used for amyloid positive and negative subjects? This precludes direct comparisons between the two groups; a single mask would be more appropriate.

7) The authors mention results are similar using a lower-resolution atlas, but the results are not shown. Please provide these supplementary results, as reproduction across atlases is important.

8) Why do the authors covary for diagnosis in voxelwise analysis? I'm not sure diagnosis is a confounding variable here, and I worry covarying for it may actually reduce effect sizes unnecessarily. Also, since covariates are (apparently) not regressed out prior to tau-accumulation covariance analyses, why is the step performed here? Given the association with age and off-target binding, are the results of the covariance analysis consistent if age is regressed from the tau-PET data prior to covariance analysis?

9) It is unclear to me how the authors executed the bootstrapping procedure to test their results against null models. Typically a permutation test (without replacement) is used to generate a null distribution, against which the test statistic is compared. The approach the authors use seems to involve random subsampling with replacement, which is more typically used for establishing confidence intervals. Also, a more realistic null model might involve shuffling the rsfMRI connectome, but preserving the degree and/or strength distribution to ensure a "brain-like" null model (see e.g. Misic et al., 2015 Neuron; Zheng...Dagher 2018 BioRxiv).

10) A word of caution regarding the "predictive" model from Figure 6: this appears to be a variation of a leave-one-out approach, which tends to be optimistically biased – see Varoquaux et al., 2017 Neuroimage.

RESULTS

11) The maps in Figure 1 produce some unexpected areas of "tau positivity", likely due to the arbitrary cutoff ascribed to the images. For within-group comparisons (both longitudinal and cross-sectional), please use one-sample t-tests and display changes that are significantly different from 0 with appropriate corrections, which will incorporate information about variability across the group instead of just the mean.

12) The analysis and results presented in Figure 5 are a bit peculiar. It seems that the authors are essentially correlating autoregressive patterns. An indication of this is that the null distribution of the test statistic ranges from 0.6-0.7, so, while the effect is indeed greater than chance, the effect size is highly inflated and a bit deceptive. I don't find this analysis adds unique information to the manuscript.

13) Please provide a table with subject demographics

14) Figure 4C is a nice summary, but I would be interested to see a similar plot to 4A plotted separately for each of the canonical networks. In other words, is the covariance in tau

accumulation better or worse correlated with functional connectivity within different large-scale networks?

15) Please provide a legend indicating what each color corresponds to for markers in Figure 5 subpanels

GENERAL

16) Please consider making the R code used to perform the analyses described in the manuscript openly available so future researchers can validate and reproduce your findings.

Reviewer #1 (Remarks to the Author):

The paper by Franzmeier and colleagues shows that non-demented individuals with amyloid pathology exhibit faster tau accumulation in temporo-parietal regions than controls, which has been shown many times before. However, the authors revealed that functionally connected regions show correlated tau accumulation rates. Further, they found that subsequent tau accumulation in pre-dementia AD could be predicted by a combination of baseline tau levels, functional connectivity and distance between brain regions. The latter findings are of great interest to the field. The paper is generally well written, but there are some major concerns:

1) The study included very few individuals (15 Ab- CU and 30 Ab+ CU/MCI) and the follow-up time is very short (appr. 1.4 y). The authors should consequently include an independent validation cohort, which have at least 2.0 y of follow-up. Further, it is unclear how the authors selected the 15+30 cases for their study, considering that there is a much larger number available in ADNI.

We agree with the reviewer that the sample size of 15 Ab- CN and 30 Ab+ CN/MCI is a limitation that may potentially impede the generalizability of our findings to larger cohorts. The reviewer is right that the ADNI database includes more individuals with tau-PET, however, most of these tau-PET data are cross-sectional, since tau-PET has been added only recently to ADNI3. The original sample size of 45 is a result of our multi-modal and longitudinal inclusion criteria (i.e. at least 2 tau-PET scans, plus availability of baseline resting-state fMRI, structural MRI and AV45 amyloid-PET), where the major limiting factor is the availability of longitudinal tau-PET. The data for the 45 subjects included in the original version of the manuscript was downloaded from the ADNI database in September 2018, but longitudinal tau-PET scans from additional subjects have been added to the ADNI database since then. To address the reviewers concern regarding the limited sample size, we thus added all subjects that met our inclusion criteria whose longitudinal tau-PET data has been uploaded since September 2018. The updated sample includes now a total of 28 Ab- CN and 53 Ab+, which is almost a doubling of the initial total sample size from originally N = 45 to now N = 81. When repeating all statistical analyses on this updated sample, all results remained fully consistent with the analyses presented in the original version of the manuscript. Regarding follow-up time, we would like to note that the follow-up time of on average 1.4 years is comparable to previous studies that have reported tau accumulations in Ab+ subjects (e.g. Jack et al., Brain, 2018, median of 1.2; Chiotis et al., Molecular Psychiatry, 2017, mean of 1.6 years; Pontecorvo et al., Brain, 2019, 1.8 years; Harrison et al., Annals Neurol, 1.3 years in Ab+ vs. 1.9 years in Ab-). Further, the Ab+ group shows significant longitudinal tau-PET increase (Figure 1 E&F), suggesting that the follow-up duration is sufficient to study tau accumulation in this group. We believe that the updated sample significantly strengthens the manuscript and sufficiently addresses the reviewers' concerns regarding sample size and thus overcomes the requirement of a second validation cohort. All analyses within the manuscript are now modified accordingly.

2) The investigated population does not include any cases with AD dementia, which is also a severe limitation, because it is vital to understand whether the spread of tau follows the same principles also during the dementia stages of the disease.

The limited availability of AD Dementia subjects is a result of the ADNI3 recruitment strategy, which focuses primarily on CN and MCI subjects (see <https://clinicaltrials.gov/ct2/show/NCT02854033>). However, following the recently published NIA-AA framework, the Ab+ subjects included in the current study fulfill the criteria of AD. Whereas the focus of the present study was on AD at the presymptomatic and early symptomatic phase, our and others' previous cross-sectional studies in AD dementia have shown that functional connectivity is also associated with tau covariance in subjects with AD dementia (Cope et al., 2018; Franzmeier et al., 2019). Therefore, we expect that the current findings generalize to AD dementia. Nevertheless, we agree with the reviewer, that future studies need to confirm this in AD dementia. Thus, we have now added a statement to the discussion section, highlighting that our findings are currently limited to early and mid-AD stages and warrant further validation across the entire AD spectrum (page 15).

3) The population of Ab- cases should be larger, and the authors should investigate whether the regional spread of tau is associated with functional connectivity, also in cases without Ab pathology (PART).

To address the reviewers concern regarding the small sample size of the CN Ab- group, we have almost doubled the N of the Ab- subjects from 15 to 28 CN-Ab- subjects (see response to Comment 1). Using this larger Ab- sample, we did not detect abnormal tau-PET levels at baseline or follow-up (Figure 1A), nor significant tau-PET accumulation across time, so PART was not detectable in the current sample of CN A β - subjects. For Ab- subjects with cognitive impairment (i.e. MCI Ab-) we found only 13 subjects to meet our inclusion criteria (i.e. baseline structural MRI and resting-state fMRI plus two AV1451 tau-PET scans). Of these subjects, only three showed significant tau-pathology (i.e. SUVR > 1.3) in inferior temporal brain regions at baseline and there was no significant tau-PET change across time as assessed by a voxel-wise t-test against zero. Thus, we believe that the Ab- sample lacks the power to study tau-spreading. Nevertheless, we agree with the reviewer that it is critical to study mechanisms of tau spread not only in AD but also in other tauopathies such as PART, since preclinical work has also described connectivity-mediated tau spreading in the absence of amyloid (Wu et al., 2016). We mention now explicitly in the conclusion of our section that future studies may use our methodological framework to study tau spreading mechanisms also in other tauopathies such as PART (p. 17).

4) When the authors include an independent validation cohort, the data could also be pooled to study

whether the prediction models differ in 1) younger vs older individuals, 2) females vs males and 3) APOE4 carriers vs non-carriers.

These are excellent suggestions. Based on the enlarged sample of 53 Ab+ subjects, we tested whether the subject-specific prediction model performance (i.e. β -values for models 1-3 as shown in Figure 6F) differed between 1) males and females, 2) subjects carrying zero, one or two ApoE4 alleles, 3) younger or older subjects. Using ANOVAs, we did not find prediction performance (i.e. β -values) to be different between males and females (Model 1: $F(51,1)=2.11$, $p=0.152$, Model 2: $F(51,1)=0.021$, $p=0.885$, Model 3: $F(51,1)=2.004$, $p=0.163$), nor associated with the number of ApoE4 alleles (Model 1: $F(50,2)=0.123$, $p=0.884$, Model 2: $F(50,2)=0.426$, $p=0.656$, Model 3: $F(50,2)=0.594$, $p=0.556$). Using linear regression, we tested the association between age and prediction performance, where we did not find a significant association between age and prediction performance (Model 1: $t(51)=-0.203$, $p=0.840$, Model 2: $t(51)=-0.074$, $p=0.941$, Model 3: $t(50)=-0.101$, $p=0.920$). These findings suggest that functional connectivity predicts the spread of tau independent of age, gender or ApoE. We thank the reviewer for this interesting idea and have added these analyses to the results section of the manuscript (p.12).

5) The authors should replicate their findings using PVEc data. Further, it is unclear to me whether baseline and follow-up PET scans were always co-registered to the MRI scans closest in time to each PET scan (i.e. the baseline and follow-up MRIs that were closest to FTP PET in time).

In the current study, all PET images were coregistered and spatially normalized using the baseline MRI scan, which was closest to the first tau-PET scan. This approach was chosen since many subjects of the current sample had only a baseline MRI scan available in the ADNI database. Thus, partial volume correction was not possible. However, we performed sharpening as described by Jack et al. (Jack et al., 2017), i.e. we included only voxels for which the likelihood of belonging to the grey matter was higher than belonging to the CSF, which has been previously shown by the Mayo Group to reduce the likelihood of baseline atrophy driving our results (Jack et al., 2018). Also, we would like to refer the reviewer to previous work (Jack et al., 2018), demonstrating comparable rates of longitudinal changes in tau-PET when comparing pve-corrected and uncorrected PET data in cognitively unimpaired and cognitively impaired AD patients (Jack et al., 2018). In a similar vein, other studies showed longitudinal voxel-wise tau increases without partial volume correction (Harrison et al., 2018; Pontecorvo et al., 2019). We would like to refer the reviewer to p.16 of the discussion section, where we have discussed this point in more detail. To address the reviewers' concern, we have now added a more in-depth discussion of missing partial volume correction to the manuscript (p.16).

6) It is very unfortunate that the authors do not include the hippocampus in their analyses, considering the central role of this structure for both brain connectivity and tau-pathology. When using PVEc data, it should not be a problem to include FTP data from hippocampus.

The reviewer is absolutely correct that the hippocampus plays a central role in the development of tau-pathology and that it acts a functional hub linking early-tau accumulating regions to the rest of the brain (Braak & Braak, 1991). The reviewer further states that including the hippocampus should be facilitated by applying partial volume correction, as suggested for instance in (Scholl et al., 2016). However, previous studies have shown significant AV1451 off-target binding in structures adjacent to the hippocampus, such as the choroid plexus (Lemoine, Leuzy, Chiotis, Rodriguez-Vieitez, & Nordberg, 2018). Off-target binding of the choroid plexus has been shown to spill over to the hippocampus even after performing partial volume correction (Lee et al., 2018). In view of these previous results, we believe that the interpretability of the AV1451 tau-PET signal in the hippocampus is limited even after partial volume correction. Since the overall analysis included 400 ROIs we also believe that the effect of one hippocampal ROI per hemisphere on the overall result pattern would be minimal. Due to these considerations, we refrained from including the hippocampus for the current study. Here, second generation tau-PET tracers with a better off-target binding profile will allow to validly test the role of the hippocampus in the spread of tau. We have now discussed this topic in more detail in the discussion section and point to second generation tau-PET tracers, which may help overcome these limitations in future studies (p.15 bottom-16).

7) It is unclear which reference region the authors used for FTP PET. I strongly recommend the authors to show whether they get similar results using either 1) inferior cerebellum, or 2) eroded white matter (PMID: 30597624) as the reference region.

We apologize for not having included information on how we intensity normalized the AV1451 tau-PET data. All tau-PET images were intensity normalized to the inferior cerebellar grey, following a previously established protocol (Baker, Maass, & Jagust, 2017). We have now added the missing information to the methods section of the manuscript (p.18 bottom-19).

Further, the reviewer asks whether our results remain consistent when using eroded white matter as the reference region. To address this, we have intensity-normalized all subjects' AV1451 tau-PET images to an eroded white matter mask following the methods described in the paper referenced by the reviewer (Harrison et al., 2018). Specifically, the eroded white matter mask was defined as the group-average white matter mask binarized at a probability threshold of 95% which was subsequently eroded by 1 voxel. When using these data to determine covariance in tau change, we find a correlation with functional connectivity ($r=0.37$, $p<0.001$) that is consistent with our main analyses using the inferior cerebellar grey as a reference region. Analogous to the plots presented in Figures 5 D&E, we also confirm that functional connectivity of the region with highest tau change

predicts high tau change in connected regions ($r=0.53$, $p<0.001$), whereas functional connectivity of the region with lowest tau change predicts low tau change in connected regions ($r=0.27$, $p<0.001$). For the prediction of longitudinal tau change (analogous to Figure 6B-D), we find that tau-weighted by distance shows the least strong association with future tau changes ($r=0.30$, $p<0.001$), whereas tau-weighted functional connectivity ($r=0.42$, $p<0.001$) and tau- and distance weighted functional connectivity ($r=0.51$, $p<0.001$) show stronger associations with future tau change. Together, this indicates that all main analyses are comparable when using the inferior cerebellum or the eroded white matter as a reference region, suggesting that the association between functional connectivity and tau accumulation is not driven by the choice of the reference region. In order to not further complicate the analyses presented within the main manuscript, we decided to restrict the presentation of these analyses to the rebuttal letter (which will be available online in case the manuscript is accepted for publication).

8) In Figs 1A-B, I am unsure how the authors established a cutoff of 1.2 SUVR when visualizing the FTP uptake. I would recommend 1.3-2.5 SUVR, which is representing a more correct FTP signal less affected by noise.

Following the reviewers' comment, we have now applied a cutoff of 1.3 SUVR when visualizing abnormal FTP uptake in Figure 1, which is similar to the cut-offs used in previous work and should remove further noise from the images (Maass et al., 2017). As a result, we do not find elevated inferior-temporal tau-PET signal in the CN Ab- anymore. Also, we have now separately plotted baseline and follow-up tau-PET data in Figure 1, to emphasize the increase of abnormal tau-PET uptake in the Ab+ group. Note that we did set the upper boundary of the color scale to 1.8 rather than 2.5 (i.e. all values above 1.8 are displayed but are shown in the same intensity in red), since this color range facilitates the discrimination of subtle tau-PET uptake differences in the CN-Ab+ who have relatively subtle tau-PET elevations.

9) In Figs 1D-E, I think the authors should present the voxels with significant change in FTP uptake compared to baseline (i.e. T-values), and not just group-average longitudinal FTP-PET changes. Panel F (not panel D as written in the legend) show the voxels exhibiting significantly different FTP PET slopes between Ab- CU and Ab+ CU/MCI, and the significant clusters are quite small and scattered indicating that the study sample is small.

To address the reviewers concern, we have now modified Figure 1: We show now for each study group (CN Ab-, CN Ab+, MCI Ab+) baseline and follow-up abnormal tau-PET uptake at a threshold of 1.3 (Figure 1 A-C). Further, we follow the reviewers' suggestion and have now assessed which voxels show significant longitudinal tau-PET changes (i.e. t-test against zero on annual changes in tau-PET) for CN Ab- and Ab+ (Figure 1 D&E). Here, we show that CN Ab- show no significant tau-PET changes across time (Figure 1D). In contrast, Ab+ subjects show significant widespread temporoparietal and some frontal tau-PET increases (Figure 1E). We have also re-assessed the group comparison between CN Ab- and Ab+ in annual tau-PET changes, showing more widespread and robust changes in Ab+ compared to CN Ab-, most likely due to the increased sample size. All Figures and the results section have been modified accordingly.

10) The association between "FC" and "covariance in tau change" is quite weak, as shown in Fig 4A. Could the authors investigate whether other factors like age, sex, APOE, education level is affecting this association.

The reviewer makes an important point, i.e. that the covariance in tau change may be affected by covariates such as age, gender, APOE or education. To address this, we have assessed the covariance in tau change between each possible ROI pair using linear regression, this time controlling for age, gender, education and ApoE load (i.e. carrying 0, 1 or 2 alleles). The standardized beta coefficient derived from these regression models should thus reflect the covariance in tau change between two regions, independent of these covariates. Using covariate-controlled covariance in tau change, we then re-assessed the association between covariance in tau change and functional connectivity. Here, we found an association of $r=0.38$, $p<0.001$ consistent with our main analysis ($r = 0.38$), suggesting that the associations between covariance in tau change and functional connectivity are not driven by differences in age, gender, education or ApoE. We thank the reviewer for this helpful comment and have included these additional analyses in the methods (p.21 & 22 bottom) and results (p.8) section of the manuscript.

11) Further, I am unsure whether the FC in Ab- CU (Fig 2B left) is used for the analyses (in Fig 4A) or the FC in Ab+ CU/MCI cases. To me it would make more sense to use the FC established in individuals without Tau pathology (i.e. Ab- CU) to avoid risk of circularity (i.e. that Tau pathology affects FC, which is then associated with change in FTP PET uptake). If the authors used the FC from Ab- CU for this comparison, they could probably use a much larger sample of Ab- CU cases from ADNI to establish regional FC. Similarly, a structural connectome could be established using DTI in ADNI, which would substantially strengthen the study.

The functional connectivity estimates of the CN Ab- has been used for illustrational purposes in Fig 2A, and for a group comparison of FC between the CN Ab- and the Ab+ subjects. Here, no significant Bonferroni-corrected ($p<0.05$) differences in FC were found between both groups, suggesting that functional connectivity is not yet disrupted in the current Ab+ sample. Nevertheless, we agree with the reviewer that finding an association between covariance in tau change in the Ab+ and functional connectivity from a healthy sample would help

address the above-mentioned risk of circularity. To this end, we have assessed the association between covariance in tau change derived from the 53 Ab+ subjects and functional connectivity derived from the healthy sample of 28 CN Ab- subjects. Here, we find a correlation of $r=0.36$, $p<0.001$, which is very similar to our main analysis using functional connectivity from the Ab+ sample ($r=0.38$, $p<0.001$). As an additional validity check, we used an equivalent 400x400 ROI connectivity matrix derived from 500 subjects from the human connectome project, i.e. an average connectivity matrix from healthy young subjects. When using this connectivity matrix, we find a correlation of $r=0.31$, $p<0.001$ with covariance in tau change, which is slightly lower but consistent with our main analysis. Thus, we are confident that our results are not specifically confounded by tau-related functional connectivity disruptions that may occur in the Ab+ group.

Second, the reviewer suggests including structural connectivity analyses to 1) assess the association between covariance in tau change and structural connectivity or 2) to combine structural and functional connectivity. While we agree that such an analysis would be of interest for the field, we believe that including DTI would go beyond the scope of the current study, which is strictly focused on functional connectivity. Specifically, including DTI would 1) further limit the size of the current sample and 2) the current standard diffusion imaging sequence included in ADNI would only allow to track the most prominent fiber tracts, whereas smaller cortico-cortical connections or crossing-fibers would be missed. Here, more technically advanced acquisition protocols like compressed sensing diffusion spectrum imaging would be desirable (Tobisch et al., 2018). Here we would like to refer the reviewer to p.16 of the manuscript, where we have discussed this point further. Since, previous findings suggest that functional and structural connectivity show similar associations with cross-sectional tau deposition (e.g. Vogel et al., BioRxiv, 2018, <https://doi.org/10.1101/555821>), we have now added an additional sentence to the discussion section, motivating future studies to test the association between structural connectivity and longitudinal tau spread (p.16).

12) Both models building on either Euclidean distance or FC were significantly associated with change in FTP PET. Could the authors test another model which they do not believe in to make sure that not all models become positively associated with spread of tau. E.g. they could study whether the covariance in FTP PET change is associated with either i) the covariance in cortical thickness or ii) blood flow (ASL).

We agree that contrasting FC vs a control condition is crucial for the validity. However, we like to point out that the comparison of effects (e.g. FC vs other predictors) based on p-values is heavily dependent on sample size, where even small effects may become significant for larger samples (i.e. across 400 ROIs). Therefore, effect size rather than p-values may be more informative. Our results showed prediction accuracy to be dramatically higher when tau spread was modeled as a function of tau-weighted functional connectivity ($R^2=22.2\%$) compared to the prediction based on Euclidian distance ($R^2=6.4\%$). Thus, although both models are significant, there are strong differences in the effect size of the predictive value, suggesting that the hypothesized predictive value of FC is clearly higher compared to the physical distance between regions.

The reviewer suggests testing either covariance in cortical thickness or in blood flow as control conditions. However, the interpretation of those analyses would not be unambiguous: First, previous covariance studies on cortical thickness and perfusion have shown that such covariance patterns resemble resting-state functional networks and may thus be a proxy of structural and functional connectivity in the brain (Tijms, Series, Willshaw, & Lawrie, 2012) (Chen, Jann, & Wang, 2015). Importantly, such covariance patterns when assessed in AD may most strongly be driven by pathology, e.g. pathologic tau, where, as the reviewer pointed out previously, the pitfall of circularity would apply. Thus, we believe that our demonstration of clear differences in the size of the predictive effect of local diffusion (Euclidean distance) vs FC in the current study makes an unambiguous and strong case in favor of FC as an important factor for predicting the spreading of pathologic tau.

13) Can the authors build a model based on their results predicting the spread through cortex of tau given different "Tau starting regions", e.g. entorhinal cortex or hippocampus? This could perhaps be nicely visualized, which would make it much easier for the reader to compare and judge the relevance of models 1, 2 and 3.

With the current approach, we try to predict the future change in tau-PET in the entire cortex based on the pattern of cortical tau-PET uptake at baseline considering functional connectivity and distance between brain regions. In principle, our model assumes that the future tau-PET increase in a given target brain region is a function of tau-PET in all other brain regions at baseline, weighted by their functional connectivity and distance to the target region (see for instance Figure 6A). Thus, the current approach does not assume one particular starting region of tau (e.g. hippocampus or entorhinal cortex), but rather takes into account the overall distribution of the tau-PET signal across the entire cortex. However, the reviewer is correct that a visualization of the prediction models can help judge the relevance of the predicted outcome. To this end, we have now mapped the actual tau-PET change as well as the predicted tau-PET change for each of the prediction models in Figure 6.

Reviewer #2 (Remarks to the Author):

From the idea that neural connectivity is related to tau accumulation and propagation, Franzmeier et al. have shown that functional connectivity may predict tau propagation in patients with Alzheimer's disease. In a cohort of 15 healthy and 30 pre-dementia Ab-positive subjects, they used baseline and longitudinal tau-PETs to follow tau accumulation, and baseline resting-state fMRI to observe tau values along functionally connected networks. Firstly, authors proved that tau levels at baseline and in longitudinal follow-up increase among Ab-positive individuals. Moreover, among the Ab-positive group, they found a significant spatial relationship between functional connectivity and tau-signal covariance, i.e. higher tau accumulation was seen in more functionally connected areas (according to baseline fMRI). Moreover, Franzmeier et al. showed that this association has predictive value, depending on tau-PET annual change. Particularly, high connectivity relates to high tau-PET change over time in specific target areas only when tau-PET change is high in seed areas, while high connectivity shows the opposite association when tau-PET change is low in seed areas. After confirming that tau spreading is not independent of functional connectivity (negative control), the group created a predictor model for tau propagation based on both functional connectivity and Euclidean distance, as they proved to be better predictors than distance or connectivity separately. Therefore, authors show that tau spreading seems to be an active process dependent on connectivity and connectivity distance.

Overall, the design and execution of this study are sound and findings are very appealing. I have a few suggestions mostly regarding the methods' section of the manuscript.

1. My main concern comes from the spatial regression analysis used in the study. I think this approach is very informative for the purposes of this investigation but authors should explain in more detail how the bootstrapping approach was accomplished. In the present version, authors just mention that the bootstrapping strategy was "iteratively determined via sampling", with no information about how this was done. If I understood well the underlying idea, authors have performed a bootstrapping approach to provide spatial regressions with the actual p-values. As this type of regression (spatial correlations) uses non-independent data (e.g. ROIs are spatially related and therefore non independent from each other), a bootstrapping strategy is necessary to build null hypothesis distributions and avoid the violation of independency between dots in the regressions. However, authors provided little detail about the bootstrapping strategy, which limits the evaluation of this critical step. Moreover, I think authors should delete the initial p-value provided in the (e. g. page 7), as these ones are not p-values derived from null hypothesis distributions.

We thank the reviewer for this important remark and would like to briefly clarify the rationale and approach of the bootstrapping procedure. We used the pool of 53 Ab+ subjects from which we iteratively drew 1000 random samples (with replacement). Then, we determined for each of these 1000 randomly drawn (i.e. bootstrapped) samples the group-average functional connectivity (i.e. 400x400 ROI FC matrix), covariance in tau change (400 x 400 matrix) and group-average tau-PET or tau-PET change (400-element vectors). Using these "iteratively determined" measures, we then tested 1) the association between functional connectivity and covariance in tau change (Figure 4), 2) the association of a given regions' tau change with tau change in connected regions (Figure 5) and 3) the performance of our 3 prediction models for future tau change (Figure 6). For each approach within each bootstrapping iteration, we saved the resulting beta values, to obtain the distribution of beta-values across these randomly selected samples. The resulting distribution of beta values and their 95% confidence interval was then plotted as a histogram. When the mean of the distribution did significantly deviate from zero (i.e. as shown by a t-test) and when the 95% Confidence interval did not include zero, we reasoned that the association between functional connectivity and tau should be robust across the sample.

Thus, we did not use p-values from individual bootstrapping iterations, nor did we use bootstrapping to build a null-distribution. Rather, we used bootstrapping to ensure that our results were consistent when randomly drawing samples from the overall pool of 53 Ab+ subjects. We have now clarified the bootstrapping procedure in more detail in the statistics (p.22) and results section (p.8) of the manuscript.

2. It is bit confusing the explanation (or implementation?) of the neuroimaging scrubbing approach to address the micro-movement. In conventional approaches, researchers either keep the interpolated volumes or remove volumes with high movement and equalize the final number of volumes among all individuals. As far as I can read in this section, authors have used different final number of volumes in each individual. Please clarify, as this may introduce a misbalance in the degrees of freedom across the sample.

We apologize for any confusion on our scrubbing approach. In brief, we marked those volumes that exceeded a framewise displacement threshold of 0.5. The marked volumes were then censored together with one preceding and two subsequent volumes, i.e. we replaced these censored volumes with zero-padded volumes which do not introduce any variance or bias in the computation of functional connectivity, but keep the number of volumes and degrees of freedom consistent across subjects. We have now clarified our scrubbing approach in more detail in the methods section (p.19).

3. Please clarify in the text (methods' section) how negative correlations were handling in the processing of the functional connectivity data.

Negative functional connectivity values have been included in all analyses and were not treated differently from positive functional connectivity values. As indicated in the group-average functional connectivity matrices (Figure 2), negative connectivity values were, however, relatively sparse and thus unlikely to drive the overall result pattern. To specifically address whether negative connectivity values biased the current results, we exploratorily thresholded all connectivity matrices at zero, retaining only positive values. Here, we found that restricting the connectivity matrices to positive values yielded consistent associations between functional connectivity and covariance in tau change ($r=0.37$, $p<0.001$). We have now mentioned more explicitly that we retained all connectivity values in the methods section of the manuscript (p.20).

4. Please rephrase the intro sentence that includes: "...has been never tested...", as this is not itself a criteria of novelty.

We have rephrased the relevant section.

5. Please replace the term "infect" in the intro.

We have rephrased the relevant section.

6. Finally, it is unclear to this reviewer how authors' interpret their "distance" results in the discussion. Is it that more Euclidean distance is positively associated to tau spreading? or less distance is positively associated to tau spreading?

The reviewer raises an important point. We assume that tau spreading is fastest across strong and spatially short connections. Thus, shorter distance should be associated with stronger tau spreading, which is in line with our main results. We mention this now explicit in the results section of the manuscript (p.10 & 11).

Reviewer #3 (Remarks to the Author):

Franzmeier and colleagues analyze longitudinal tau-PET data in a small sample of non-demented individuals from the ADNI3 dataset, and compare whole-brain regional covariance in tau accumulation with whole-brain regional functional connectivity. Building on their recent publication using cross-sectional data (Franzmeier et al., 2019 Brain), the authors show that regional covariance patterns in longitudinal tau-PET accumulation also correlates with regional functional connectivity, adding to increasing evidence that tau pathology may be transmitted through synaptic connections in human Alzheimer's disease.

This is a compact and focused study with a good introduction, concerning a timely topic of great importance to the field of Alzheimer's disease and dementia. However, it is not clear to me that the results presented here provide a substantial advance over the results of Franzmeier et al., 2019 Brain. While finding similar results using longitudinal data is encouraging, I have additional concerns regarding certain methodological choices and incomplete reporting of crucial methodological details. Please find detailed list of my concerns below:

1) My biggest conceptual concern is that the results presented here do not differ much from Franzmeier et al., 2019 Brain. While longitudinal data is introduced, the sample size is smaller and far less representative of the Alzheimer's disease spectrum, representing only the earliest manifestation of the disease. The cross-sectional and longitudinal patterns from Fig. 1 are largely consistent with Harrison et al., 2019; Jack et al., 2018, Pontecorvo et al. 2019, and the main results are highly consistent with Franzmeier et al. 2019. While a spreading model is introduced in Figure 6, this model is well behind the state of the art of spreading models, such as those presented in Raj et al. 2012 Neuron; Iturria-Medina et al. 2014 PLoS Comp. Biol; Misisic et al., 2015 Neuron; Zheng...Dagher 2018 BioRxiv; Olivier...Fleming 2019 BioRxiv; etc. All in all, it is not clear that the addition of longitudinal data adds much new information about the relationship between tau spreading and connectivity in humans.

We thank the reviewer for this critical remark. To address the reviewer's first concern regarding the limited sample size, we would like to highlight that we have almost doubled the sample size to now 81 subjects (53 Ab+/28 CN Ab-), compared to the initial sample of 45 (30 Ab+/15 CN Ab-). Specifically, we have now included longitudinal tau-PET data that has been added since the initial dataset for the current study has been downloaded in September 2018 (see also our response on comment 1 by reviewer 1). All statistical analyses reported in the original manuscript have been repeated with the now larger sample, yielding fully consistent results to those that we reported initially, supporting the robustness of our findings. All figures and analyses have been modified using this larger sample.

Second, the reviewer notes that the high consistency of the current findings with our previous work (Franzmeier et al., 2019) limits the novelty of the current study. As the reviewer correctly points out, the previous work by us and others showing an association between functional connectivity and tau-PET has been purely based on cross-sectional data, which was the key limitation of these studies (Cope et al., 2018; Franzmeier et al., 2019; Hoenig et al., 2018). Specifically, a cross-sectional study design does not allow studying a process like tau spreading that can by definition only be observed longitudinally. Thus, the only critical conclusion that could be drawn from these previous studies was that the pattern of tau-PET is spatially similar to functional brain networks. However, whether connectivity is related to future tau accumulation in AD patients, indicative of connectivity-mediated spread of tau, remained unclear. So far, the association between connectivity and tau accumulation could only be studied in preclinical studies using transgenic animals or cell cultures, where studies have shown that structural connectivity between brain regions and shared neuronal activity (i.e. functional connectivity) promote spreading and accumulation of tau (Wu et al., 2016). Translating these findings to data in living AD patients is thus a critical validation of a concept that is already used as a target for therapy (e.g. the BIIB092 antibody currently tested in a Phase II clinical trial, targets extracellular tau, which is thought to be directly involved in tau spreading). Thus, we believe that the current study makes an important contribution in the understanding of tau spreading in vivo.

METHODS

2) The authors do not provide sufficient detail in their PET preprocessing methods to properly evaluate their longitudinal pipeline. The authors for example do not mention which reference region they use, a topic of controversy for longitudinal tau-PET (see e.g. Harrison et al. 2019). How were the reference regions derived (native space or template space)? Does each subject only have one structural MRI? Are T1s segmented or masked before spatial normalization? Is the tau-PET data smoothed? Please provide full details of the methodological pipeline

We thank the reviewer for these important remarks and would like to clarify our approach in more detail. The reference region we used for intensity normalization of the tau-PET data was the inferior cerebellar grey, which was determined on baseline structural MRI scans in subject space, following a previously described pipeline (Baker et al., 2017; Maass et al., 2017). We chose the inferior cerebellar grey instead of the whole cerebellum, since previous work has shown tau-PET binding in the superior cerebellum, which may confound SUVR estimation when considered for computing SUVRs (Baker et al., 2017; Maass et al., 2017). Please note that reviewer 1 asked for validation of our results using an alternative reference region (eroded white matter) that has been used by previous longitudinal tau-PET studies (Harrison et al., 2018). Here, all analyses remained fully consistent with the results presented in the revised version of the manuscript. We would kindly refer the reviewer to our response on comment 7 by reviewer 1 for more details on these analyses.

Due to the limited number of available longitudinal structural MRI scans, only the baseline structural MRI scans were used for coregistration and spatial normalization of the tau-PET data. Structural MRI scans were skull-stripped before estimating the non-linear transformation parameters to MNI space via ANTs. For ROI based analyses, no smoothing was applied to the tau-PET images in order to avoid spill over between adjacent ROIs which may artificially introduce covariance in tau change between spatially adjacent regions (Franzmeier et al., 2019). For voxel-wise comparisons shown in Figure 1, however, we applied an 8mm FWHM smoothing kernel to grey matter masked and spatially normalized tau-PET images. We have now clarified the processing pipeline in the methods section of the manuscript (p.18, 19 & 21).

3) The small sample size remains an issue, but may be partially abated by not restricting the sample to subjects with rsfMRI scans. These functional scans are hardly used at the individual level anyway, and can be substituted with a template connectome based on other ADNI subjects and/or other datasets. Certainly more longitudinal ADNI tau scans are currently available than are included in the present analysis.

The limited sample size was also a major concern raised by reviewer 1 (see also our response on comment 1 by reviewer 1). To address this, the reviewer suggests using longitudinal tau-PET data from ADNI in combination with resting-state fMRI data from a normative sample. We appreciate this suggestion, which would certainly help to include more subjects. However, using connectivity and tau-PET data from different individuals is also a limitation as no 1 to 1 matching of the tau-PET and connectivity data is possible, which is done for instance during the bootstrapping procedures (see our response on comment 1 by reviewer 2) or while estimating the subject-specific prediction models. Thus, we decided to increase the sample size by maintaining our inclusion criteria and adding those ADNI subjects for which longitudinal tau-PET data has been uploaded since September 2018 (i.e. when the data for the originally submitted version was downloaded and processed). This step has led to a drastically increased overall N of 81 subjects (53 Ab+ vs. 28 CN Ab-) in comparison to the overall 45 (30 Ab+ vs. 15 CN Ab-) subjects that were used in the original manuscript. When using the updated sample, all results reported within the manuscript remained consistent with the results of the originally submitted manuscript. We believe that this increase in sample size substantially strengthens the results of the current study and sufficiently addresses the initial limitation of the low sample size. We would also like to refer the reviewer to our response on comment 11 by reviewer 1, where we discuss a related topic.

4) The authors mention using a Spearman's correlation to compute regional covariance in tau accumulation, an appropriate choice given the sample size and data. However, these values are then Fisher's r-to-z transformed, which will pull out the tails of the r distribution, essentially achieving the opposite goal of the Spearman's correlation by upweighting more extreme values. How do the authors reconcile these contradicting approaches?

As the reviewer correctly points out, the rationale to use Spearman correlation for computing covariance in tau change was to minimize the influence of single outliers/extreme values in the computation of any ROI to ROI covariance in tau change, especially given the initially small sample size. While the subsequent Fisher-z transformation upweights high correlation values in the overall distribution of R-values, the initial usage of Spearman correlation should guard against any outlier driving the underlying ROI-to-ROI correlations which would otherwise be upweighted. Also, the increase of the Ab+ sample from 30 to 53 should yield more stable correlation coefficients that are less likely to be driven by single outliers. Still, we agree with the reviewer that it is reasonable to assess whether the Fisher-z transformation step may bias our results. When testing the association between functional connectivity and covariance in tau-change (as raw Spearman correlations), we find an r of 0.35, $p < 0.001$, which is consistent with our main analysis using Fisher-z transformed correlations (i.e. $r = 0.38$, $p < 0.001$). Together, these analyses suggest that our findings are not artificially driven by applying Fisher-z transformation.

5) The fMRI motion correction seems a bit too minimalist, particularly given the clinical cohort and the lack of global signal regression. The authors do not, for example, regress motion parameters out of the time series. This may seem like a small point, but with two low-resolution/smooth measurements, there is a high risk that autocorrelation (caused by e.g. motion) may be driving correlations.

We thank the reviewer for this important remark. In line with all our previous work (e.g. (Franzmeier et al., 2019)), we regressed out the 6 motion parameters (3 translations, 3 rotations) from the resting-state fMRI images, which is a standard step in our preprocessing pipeline. We apologize for having missed this important information in the submitted version of the manuscript. We have now added this important information to the methods section of the manuscript (p.19). Together, our pipeline includes both motion regression as well as scrubbing, which should together address motion correction. Note that we did not include global signal regression due to the ongoing debate on the validity of this preprocessing step (Murphy, Birn, Handwerker, Jones, & Bandettini, 2009; Murphy & Fox, 2016).

6) Why are different brain masks used for amyloid positive and negative subjects? This precludes direct comparisons between the two groups; a single mask would be more appropriate.

The reviewer is absolutely correct that using different brain masks would preclude direct group-comparisons. In fact, we have only used a single mask across the entire cohort of CN Ab- and Ab subjects. We apologize for any confusion caused by our wording in the submitted manuscript version. We have now changed our wording of the

methods section to “Prior to all analyses, the Schaefer fMRI atlas was masked with a sample derived grey matter mask”.

7) The authors mention results are similar using a lower-resolution atlas, but the results are not shown. Please provide these supplementary results, as reproduction across atlases is important.

We agree that actually showing these replication results is critical, hence we have added all relevant Figures using the 200 ROI parcellation to the supplementary.

8) Why do the authors covary for diagnosis in voxelwise analysis? I'm not sure diagnosis is a confounding variable here, and I worry covarying for it may actually reduce effect sizes unnecessarily. Also, since covariates are (apparently) not regressed out prior to tau-accumulation covariance analyses, why is the step performed here? Given the association with age and off-target binding, are the results of the covariance analysis consistent if age is regressed from the tau-PET data prior to covariance analysis?

We appreciate the reviewers' suggestion. We have removed diagnosis as a covariate from our voxel-wise analyses. In these updated analyses, we find consistent results with those reported in the original manuscript, where the Ab+ group (i.e. CN and MCI pooled) shows stronger longitudinal tau-PET changes than the CN Ab-group especially in temporoparietal regions. Second, the reviewer asks whether controlling the assessment of covariance in tau change for covariates (e.g. age) yields consistent results. A similar point was raised by reviewer 1 in comment 10. To address this, we have assessed the ROI-to-ROI covariance in tau change controlling for common covariates including age, gender, education and ApoE status (i.e. carrying 0, 1 or 2 alleles), using linear regression. The standardized beta coefficient derived from these regression models should thus reflect the ROI-to-ROI covariance in tau change, independent of age, gender, education and ApoE. Using these covariate-corrected values, we found an equivalently strong association of $r=0.38$, $p<0.001$ (compared to $r=0.38$, $p<0.001$ without covariate correction), suggesting that the associations between covariance in tau change and functional connectivity are not specifically driven by differences in age, gender, education or ApoE. We have added this analysis to the results section of the manuscript.

9) It is unclear to me how the authors executed the bootstrapping procedure to test their results against null models. Typically a permutation test (without replacement) is used to generate a null distribution, against which the test statistic is compared. The approach the authors use seems to involve random subsampling with replacement, which is more typically used for establishing confidence intervals. Also, a more realistic null model might involve shuffling the rsfMRI connectome, but preserving the degree and/or strength distribution to ensure a “brain-like” null model (see e.g. Misic et al., 2015 Neuron; Zheng...Dagher 2018 BioRxiv).

Reviewer 2 has raised the same point in comment 1, hence we would kindly ask the reviewer to refer to our response on this comment for an in-depth clarification of our bootstrapping approach. Further, the reviewer suggests a permutation approach to generate a null distribution against which we could compare the test statistic. Following this approach, we have tested association between functional connectivity and covariance in tau change for 1000 permutations during which we shuffled the covariance in tau change values. The resulting t-value distribution (mean = 0.06, sd = 1.01, min = -2.84, max = 3.67) was then compared against the actual test statistic (i.e. association between functional connectivity and covariance in tau change: $t = 164.15$), where the resulting exact p-value was <0.0001 in line with our main analysis. This suggests that our results remain consistent when tested against a null model distribution.

10) A word of caution regarding the “predictive” model from Figure 6: this appears to be a variation of a leave-one-out approach, which tends to be optimistically biased – see Varoquaux et al., 2017 Neuroimage.

To our knowledge and based on the Paper by Varoquaux (Varoquaux et al., 2017), the term leave-one-out stems mostly from cross-validation in machine learning, where a given model is trained on a subsample of an available data pool and is subsequently tested on a held out subject or trial that has not been included in model training. Within our models, tau change in a given target ROI is predicted by tau in the remaining seed ROIs weighted either by distance (Model 1), functional connectivity (Model 2) or a combination of distance and functional connectivity (Model 3) in a linear regression framework. These input weights are determined for each ROI and tested as a predictor of actual future tau change across the entire cortex. A similarity between a leave-one-out approach and our predictive models is that tau change of a given target ROI (i.e. the “held-out” ROI) is estimated based on baseline tau of within seed ROIs weighted by distance and/or connectivity. However, in contrast to the leave-one-out approach described in (Varoquaux et al., 2017) our model does not include model training in a given set of data (i.e. training the prediction of tau-change in a given ROI set) with subsequent validation in held out data (i.e. testing the prediction of tau-change in a given ROI set), which has been shown to yield biased estimated of prediction performance. However, we see the reviewers point, that our current approach renders our models rather associative than predictive, since we correlate estimated tau change with actual tau change, so no actual “prediction” is made. We have thus changed our terminology from predictive to “associative” for the models presented in Figure 6.

RESULTS

11) The maps in Figure 1 produce some unexpected areas of “tau positivity”, likely due to the arbitrary

cutoff ascribed to the images. For within-group comparisons (both longitudinal and cross-sectional), please use one-sample t-tests and display changes that are significantly different from 0 with appropriate corrections, which will incorporate information about variability across the group instead of just the mean.

Following the reviewers' comment, we have now applied a more restrictive abnormality threshold of 1.3, which was also requested by reviewer 1 (see our response on comment 8 by reviewer 1). Further, we followed the reviewers' suggestion and have used voxel-wise one-sample t-tests to determine which brain regions show significant tau change in both CN Ab- and Ab+. Here, we find that the CN Ab- group shows no significant tau change across time, whereas the Ab+ group shows longitudinal increase in tau, especially in temporoparietal brain regions. We have added these data to the results section of the manuscript (p.6-7) and have revised Figure 1.

12) The analysis and results presented in Figure 5 are a bit peculiar. It seems that the authors are essentially correlating autoregressive patterns. An indication of this is that the null distribution of the test statistic ranges from 0.6-0.7, so, while the effect is indeed greater than chance, the effect size is highly inflated and a bit deceptive. I don't find this analysis adds unique information to the manuscript.

The main purpose of the analysis presented in Figure 5 is to emphasize that regions with high changes in tau are preferentially connected to other regions with high tau changes (e.g. as shown in Figure 5E) whereas regions with low tau changes are mostly connected to other regions with low tau changes (Figure 5D). Rather than showing this association only for the region with the highest (Figure 5E) and lowest (Figure 5D) change, we preferred to emphasize this association across the entire spectrum from low to high tau-PET changes (i.e. Figure 5A, x-axis). To avoid including a single scatterplot for each ROI in the supplementary, we decided to show the association between tau-change (x-axis) and regression-derived predictions of tau change in connected regions (y-axis) in a more comprehensive manner. Since the analysis shows that the association between seed-based connectivity and tau change in connected regions varies as a function of the seed regions own tau change, we believe that this analysis indeed adds unique information to the manuscript.

13) Please provide a table with subject demographics

Subject demographics are presented in table 1 at the end of the manuscript (referenced in the results section), which the reviewer may have missed.

14) Figure 4C is a nice summary, but I would be interested to see a similar plot to 4A plotted separately for each of the canonical networks. In other words, is the covariance in tau accumulation better or worse correlated with functional connectivity within different large-scale networks?

This is an excellent suggestion. Following the reviewers' comment, we have assessed within each of the 7 canonical networks the association between functional connectivity and covariance in tau change. Here, we find consistent and high correlations for all networks (DAN, beta=0.46; DMN, beta=0.51; FPCN, beta=0.56; Limbic, beta=0.56; Motor, beta=0.64; VAN, beta=0.50; Visual, beta=0.59, all $p < 0.001$). This analysis nicely illustrates that the association between functional connectivity and covariance in tau change is consistent across all networks tested. We thank the reviewer for this comment and have added this analysis to the results section and Figure 4.

15) Please provide a legend indicating what each color corresponds to for markers in Figure 5 subpanels

A color legend has now been added to Figure 5.

GENERAL

16) Please consider making the R code used to perform the analyses described in the manuscript openly available so future researchers can validate and reproduce your findings.

We agree that making the code publicly available facilitates reproducibility by other researchers. Thus, we have upload a template version of our code as a supplementary.

References:

- Baker, S. L., Maass, A., & Jagust, W. J. (2017). Considerations and code for partial volume correcting [(18)F]-AV-1451 tau PET data. *Data Brief, 15*, 648-657. doi:10.1016/j.dib.2017.10.024
- Braak, H., & Braak, E. (1991). Neuropathological staging of Alzheimer-related changes. *Acta Neuropathol, 82*(4), 239-259.
- Chen, J. J., Jann, K., & Wang, D. J. (2015). Characterizing Resting-State Brain Function Using Arterial Spin Labeling. *Brain Connect, 5*(9), 527-542. doi:10.1089/brain.2015.0344
- Cope, T. E., Rittman, T., Borchert, R. J., Jones, P. S., Vatansever, D., Allinson, K., . . . Rowe, J. B. (2018). Tau burden and the functional connectome in Alzheimer's disease and progressive supranuclear palsy. *Brain, 141*(2), 550-567. doi:10.1093/brain/awx347
- Franzmeier, N., Rubinski, A., Neitzel, J., Kim, Y., Damm, A., Na, D. L., . . . Alzheimer's Disease Neuroimaging, I. (2019). Functional connectivity associated with tau levels in ageing, Alzheimer's, and small vessel disease. *Brain*. doi:10.1093/brain/awz026
- Harrison, T. M., La Joie, R., Maass, A., Baker, S. L., Swinnerton, K., Fenton, L., . . . Jagust, W. J. (2018). Longitudinal tau accumulation and atrophy in aging and Alzheimer's disease. *Ann Neurol*. doi:10.1002/ana.25406
- Hoening, M. C., Bischof, G. N., Seemiller, J., Hammes, J., Kukolja, J., Onur, O. A., . . . Drzezga, A. (2018). Networks of tau distribution in Alzheimer's disease. *Brain, 141*(2), 568-581. doi:10.1093/brain/awx353
- Jack, C. R., Jr., Wiste, H. J., Schwarz, C. G., Lowe, V. J., Senjem, M. L., Vemuri, P., . . . Petersen, R. C. (2018). Longitudinal tau PET in ageing and Alzheimer's disease. *Brain, 141*(5), 1517-1528. doi:10.1093/brain/awy059
- Jack, C. R., Jr., Wiste, H. J., Weigand, S. D., Therneau, T. M., Lowe, V. J., Knopman, D. S., . . . Petersen, R. C. (2017). Defining imaging biomarker cut points for brain aging and Alzheimer's disease. *Alzheimers Dement, 13*(3), 205-216. doi:10.1016/j.jalz.2016.08.005
- Lee, C. M., Jacobs, H. I. L., Marquie, M., Becker, J. A., Andrea, N. V., Jin, D. S., . . . Johnson, K. A. (2018). 18F-Flortaucipir Binding in Choroid Plexus: Related to Race and Hippocampus Signal. *J Alzheimers Dis, 62*(4), 1691-1702. doi:10.3233/JAD-170840
- Lemoine, L., Leuzy, A., Chiotis, K., Rodriguez-Vieitez, E., & Nordberg, A. (2018). Tau positron emission tomography imaging in tauopathies: The added hurdle of off-target binding. *Alzheimers Dement (Amst), 10*, 232-236. doi:10.1016/j.dadm.2018.01.007
- Maass, A., Landau, S., Baker, S. L., Horng, A., Lockhart, S. N., La Joie, R., . . . Alzheimer's Disease Neuroimaging, I. (2017). Comparison of multiple tau-PET measures as biomarkers in aging and Alzheimer's disease. *Neuroimage, 157*, 448-463. doi:10.1016/j.neuroimage.2017.05.058
- Murphy, K., Birn, R. M., Handwerker, D. A., Jones, T. B., & Bandettini, P. A. (2009). The impact of global signal regression on resting state correlations: are anti-correlated networks introduced? *Neuroimage, 44*(3), 893-905. doi:10.1016/j.neuroimage.2008.09.036
- Murphy, K., & Fox, M. D. (2016). Towards a Consensus Regarding Global Signal Regression for Resting State Functional Connectivity MRI. *Neuroimage*. doi:10.1016/j.neuroimage.2016.11.052
- Pontecorvo, M. J., Devous, M. D., Kennedy, I., Navitsky, M., Lu, M., Galante, N., . . . Mintun, M. A. (2019). A multicentre longitudinal study of flortaucipir (18F) in normal ageing, mild cognitive impairment and Alzheimer's disease dementia. *Brain, 142*(6), 1723-1735. doi:10.1093/brain/awz090

- Scholl, M., Lockhart, S. N., Schonhaut, D. R., O'Neil, J. P., Janabi, M., Ossenkoppele, R., . . . Jagust, W. J. (2016). PET Imaging of Tau Deposition in the Aging Human Brain. *Neuron*, *89*(5), 971-982. doi:10.1016/j.neuron.2016.01.028
- Tijms, B. M., Series, P., Willshaw, D. J., & Lawrie, S. M. (2012). Similarity-based extraction of individual networks from gray matter MRI scans. *Cereb Cortex*, *22*(7), 1530-1541. doi:10.1093/cercor/bhr221
- Tobisch, A., Stirnberg, R., Harms, R. L., Schultz, T., Roebroek, A., Breteler, M. M. B., & Stocker, T. (2018). Compressed Sensing Diffusion Spectrum Imaging for Accelerated Diffusion Microstructure MRI in Long-Term Population Imaging. *Front Neurosci*, *12*, 650. doi:10.3389/fnins.2018.00650
- Varoquaux, G., Raamana, P. R., Engemann, D. A., Hoyos-Idrobo, A., Schwartz, Y., & Thirion, B. (2017). Assessing and tuning brain decoders: Cross-validation, caveats, and guidelines. *Neuroimage*, *145*(Pt B), 166-179. doi:10.1016/j.neuroimage.2016.10.038
- Wu, J. W., Hussaini, S. A., Bastille, I. M., Rodriguez, G. A., Mrejeru, A., Rilett, K., . . . Duff, K. E. (2016). Neuronal activity enhances tau propagation and tau pathology in vivo. *Nat Neurosci*, *19*(8), 1085-1092. doi:10.1038/nn.4328

Reviewers' comments:

Reviewer #1 (Remarks to the Author):

The paper has been improved. However, the lack of an independent validation cohort and lack of cases with AD dementia are still important limitations.

I am still concerned with the use of baseline MRIs to define the ROIs used for ROI-based quantification of tracer retention of both baseline and follow-up FTP PET images. I agree that PVEc might not be necessary, but PVEc is mainly correcting for the effects caused by the relatively low spatial resolution of PET and NOT change over time in a brain structure volume. The current analyses are not adjusting for differences in individual atrophy rates. Consequently, it could be the case that regional atrophy over time is associated with baseline functional connectivity independent of tau pathology. Not adjusting the follow-up MRI-defined ROIs used for quantifying FTP uptake at follow-up could consequently bias the results. I would therefore urge the authors to do a sensitivity analyses only including cases with both baseline and follow-up MRI scans close in time to the baseline and follow-up FTP PET images, respectively, and only use follow-up MRIs for co-registration of follow-up FTP PETs.

Reviewer #2 (Remarks to the Author):

The authors have nicely addressed my previous concerns. I don't have further comments. Congratulations.

Reviewer #3 (Remarks to the Author):

The revisions have substantially enhanced the quality of the manuscript. In particular, the more clear detailing of statistical approaches have inspired more confidence in the results, and additions made to the Figures and analyses have lead to more interesting findings. The authors should be commended for deciding to include the code so that their analyses can be reproduced.

There are, however, two points that remain problematic in direct relation to previous comments that still have not been appropriately addressed. While these points may seem minor, they are in fact important both for interpretation of the current results, and for others looking to perform similar analyses inspired by this manuscript.

1. The authors have done well to further clarify their bootstrapping approach to validate their statistics and it is now much more clear to me what their goals were and how they were accomplished. However, while this approach inspires some confidence, it is not a null model. The bootstrapping approach helps build confidence that the results are not driven by the sample, however a null model should ensure the results are not driven by the design. In other words, it is important to demonstrate that the results are truly driven by the specific structure of the human brain connectivity (or tau covariance) data, not the statistical properties of connectivity data in general.

In order to demonstrate this, as with my previous comments, I would encourage the authors to rerun their models across several scrambled connectomes (making sure to preserve degree and strength distributions). You can see e.g. Misić et al., 2015 Neuron; Zheng...Dagher 2018 BioRxiv as examples, and there are plenty of open tools that will help you do this (e.g. Brain Connectivity Toolbox). The "true" beta value can then be tested against the null distribution of beta values produced by running the samples across these null models, in order to derive an empirical, model specific p-value. It is important that the authors perform null models for each of their statistical

models, i.e. those presented in Figures 4, 5 and 6, respectively. This will be helpful not only in understanding whether the model could occur by chance simply based on the data structure (a common issue with highly autocorrelated data), but also in learning the range of the null distribution of such a model.

2. This may seem like a small point but it is nonetheless important. Applying a single meaningful SUVR "cutoff" to the entire brain is simply not appropriate with AV1451 SUVR data. Other groups have published with global cutoffs, but this was in the early days of AV1451-PET, under assumptions that it behaves the same as other tracers, and in lieu of a better way to describe the data. We now know that the distribution of AV1451 values vary widely by region, and a single SUVR cutoff is therefore inappropriate. This is why I previously encouraged a one-sample t-test, which the authors have applied to the longitudinal data in Figure 1. However, Figure 1 still contains cross-sectional data based on a non-empirical cutoff of 1.3. In order to avoid negatively influencing the field with what will surely be a widely read paper, please either apply a one-sample t-test or some other more empirical threshold to these cross-sectional images, or simply show the data without a threshold.

3. As a minor point, the authors may want to mention in the limitations that the use of only a single MRI for two time points of data may expose them to the asymmetry bias (Reuter & Fischl, 2011, Neuroimage).

Reviewers' comments:

Reviewer #1 (Remarks to the Author):

1. The paper has been improved. However, the lack of an independent validation cohort and lack of cases with AD dementia are still important limitations.

Response: Following the reviewer's comment, we have included an independent sample with longitudinal tau-PET data (BioFINDER, Lund University, PI Oskar Hansson). The sample includes 16 A β - cognitively normal (CN) controls, 16 CN A β +, 7 A β + subjects with mild cognitive impairment (MCI) and 18 patients with AD dementia (i.e. A β +) with a mean follow-up time of 1.92 +/- 0.36 years. As in ADNI, all tau-PET data in this sample were acquired using AV1451. Since resting-state fMRI was unavailable in these subjects, we used functional connectivity data from 500 subjects of the human connectome project (HCP) as a connectivity template. Note, that we have used these HCP data also during the previous round of revisions to confirm that the association between functional connectivity and covariance in tau change is not specifically driven by the ADNI connectivity data. Using the BioFINDER data in combination with the HCP functional connectivity data, we repeated all analyses reported in ADNI (except from the bootstrapping approach which requires combined sampling of subject level tau-PET and functional connectivity data). In the 41 A β + subjects of the BioFINDER sample, we determined the association between functional connectivity and covariance in tau change, where we found a significant association of standardized $\beta=0.30$, $p<0.001$, consistent with the association that we reported in ADNI ($\beta=0.38$, $p<0.001$). Functional network-specific analyses confirmed the association between functional connectivity and covariance in tau change across all 7 major functional networks included in the brain parcellation. Next, we could confirm that connectivity of the region with highest tau-change predicted higher tau change in other regions ($\beta=0.398$, $p<0.001$), whereas higher connectivity of the region with lowest tau change predicted lower tau change in other regions ($\beta=-0.397$, $p<0.001$). Lastly, we validated the prediction models that were introduced in the previous version of the manuscript. In line with the findings from ADNI, we find in BioFINDER that tau-weighted connectivity ($\beta=0.400$, $p<0.001$) as well as tau- and distance-weighted connectivity ($\beta=0.421$, $p<0.001$) significantly predict group-average tau change. Congruent effects were found for subject-level prediction of tau-PET changes as shown in Figure 7 of the manuscript. We have included all analyses in the revised version of the manuscript and have modified the manuscript and all figures (all changes are highlighted in yellow throughout the manuscript). We would further like to highlight that all findings in BioFINDER remain stable when using the 200 ROI instead of the 400 ROI-atlas version as shown in supplementary figures 1-5. Also, as in ADNI, exploratorily using eroded white matter instead of the inferior cerebellar grey as a reference region yielded consistent results in BioFINDER.

2. I am still concerned with the use of baseline MRIs to define the ROIs used for ROI-based quantification of tracer retention of both baseline and follow-up FTP PET images. I agree that PVEc might not be necessary, but PVEc is mainly correcting for the effects caused by the relatively low spatial resolution of PET and NOT change over time in a brain structure volume. The current analyses are not adjusting for differences in individual atrophy rates. Consequently, it could be the case that regional atrophy over time is associated with baseline functional connectivity independent of tau pathology. Not adjusting the follow-up MRI-defined ROIs used for quantifying FTP uptake at follow-up could consequently bias the results. I would therefore urge the authors to do a sensitivity analyses only including cases with both baseline and follow-up MRI scans close in time to the baseline and follow-up FTP PET images, respectively, and only use follow-up MRIs for co-registration of follow-up FTP PETs.

Response: We followed the reviewers' advice and conducted a sensitivity analysis in ADNI including only those 22 A β + subjects who had structural MRIs at the same visit as the follow-up tau-PET scan. In these 22 A β + subjects we used these follow-up structural MRI scans to co-register and spatially normalize baseline and follow-up tau-PET images. When repeating the analyses using the newly co-registered data of this subsample, we find consistent results with the ADNI analyses reported in the main manuscript. Specifically, we find that functional connectivity is correlated with covariance in tau-PET change ($\beta=0.34$, $p<0.001$). In a similar vein, higher connectivity of the region with highest tau-change predicted higher tau change in other regions ($\beta=0.568$, $p<0.001$), whereas higher connectivity of the region with lowest tau change predicted lower tau change in other regions ($\beta=-0.272$, $p<0.001$). Lastly, we tested the prediction models of longitudinal tau spread in the 22 subjects with follow-up MRI. Here, we find congruent results with our main analyses, where tau- & distance weighted connectivity are the best predictors of future tau change (i.e. $\beta=0.44$, $p<0.001$), followed by tau-weighted connectivity ($\beta=0.4$, $p<0.001$) and tau weighted

by distance ($\beta=0.13$, $p<0.001$). Together, these analyses using follow-up structural MRI data indicate that registering the PET images to the baseline MRI does not drive our results in a specific direction.

To further address the reviewers concern, we have conducted additional analyses using partial volume corrected data (Gaussian Transfer Method) that is available in the BioFINDER cohort. In line with our main results using non-partial volume corrected data, we find a significant association between functional connectivity and covariance in tau-PET change ($\beta=0.25$, $p<0.001$). Also, higher connectivity of the region with highest tau-change predicted higher tau change in other regions ($\beta=0.344$, $p<0.001$), whereas higher connectivity of the region with lowest tau change predicted lower tau change in other regions ($\beta=-0.339$, $p<0.001$). Congruent with our main analyses, tau-weighted connectivity ($\beta=0.225$, $p<0.001$) as well as tau- and distance-weighted connectivity ($\beta=0.243$, $p<0.001$) significantly predict group-average tau change.

Note that we have not included these results in the main manuscript in view of the already complex set of analyses. In case the manuscript is accepted for publication, the rebuttal letter and hence the results of these additional analyses will be publicly available on the Nature Communications homepage.

Reviewer #2 (Remarks to the Author):

**1. The authors have nicely addressed my previous concerns. I don't have further comments.
Congratulations.**

We thank the reviewer for these kind remarks

Reviewer #3 (Remarks to the Author):

The revisions have substantially enhanced the quality of the manuscript. In particular, the more clear detailing of statistical approaches have inspired more confidence in the results, and additions made to the Figures and analyses have lead to more interesting findings. The authors should be commended for deciding to include the code so that their analyses can be reproduced.

There are, however, two points that remain problematic in direct relation to previous comments that still have not been appropriately addressed. While these points may seem minor, they are in fact important both for interpretation of the current results, and for others looking to perform similar analyses inspired by this manuscript.

1. The authors have done well to further clarify their bootstrapping approach to validate their statistics and it is now much more clear to me what their goals were and how they were accomplished. However, while this approach inspires some confidence, it is not a null model. The bootstrapping approach helps build confidence that the results are not driven by the sample, however a null model should ensure the results are not driven by the design. In other words, it is important to demonstrate that the results are truly driven by the specific structure of the human brain connectivity (or tau covariance) data, not the statistical properties of connectivity data in general.

In order to demonstrate this, as with my previous comments, I would encourage the authors to rerun their models across several scrambled connectomes (making sure to preserve degree and strength distributions). You can see e.g. Misisic et al., 2015 Neuron; Zheng...Dagher 2018 BioRxiv as examples, and there are plenty of open tools that will help you do this (e.g. Brain Connectivity Toolbox). The "true" beta value can then be tested against the null distribution of beta values produced by running the samples across these null models, in order to derive an empirical, model specific p-value. It is important that the authors perform null models for each of their statistical models, i.e. those presented in Figures 4, 5 and 6, respectively. This will be helpful not only in understanding whether the model could occur by chance simply based on the data structure (a common issue with highly autocorrelated data), but also in learning the range of the null distribution of such a model.

Response: We appreciate the reviewers' concern that the paper could be strengthened by including null-models of functional connectivity as a negative control analysis. Following the

reviewers' comment, we have obtained 200 scrambled connectomes with preserved degree, strength and weight distribution using the `null_model_und_sign` function from the brain connectivity toolbox. Connectomes were scrambled for ADNI group-average functional connectivity data and for group-average functional connectivity from the 500 human connectome project subjects that was used as a connectivity template for the newly added BioFINDER sample. As suggested by the reviewer, these 200 scrambled connectomes were subsequently used to obtain null model-based beta value distributions for the analyses presented in Figures 4-6 of the previous version of the manuscript (now Figures 4, 6 & 7). For our first analysis on the association between functional connectivity and covariance in tau change, we tested for each of the scrambled connectomes, whether scrambled functional connectivity predicted covariance in tau change (i.e. analysis presented in Figure 4A&B). In ADNI, the resulting β -value distribution has a mean/SD of 0.09/0.002 and a mean/SD of -0.02/0.002 in BioFINDER. When comparing the true β -values (ADNI: $\beta = 0.38$, BioFINDER: $\beta = 0.30$) with this null-model distribution using an exact test (i.e. the percentage of scrambled connectome β -values that is greater than the true β -value), we obtain a p-value < 0.001 . In a similar vein, we repeated the analyses presented in Figure 6B&E (i.e. testing the association between seed-based functional connectivity and tau-PET change in connected regions) using 200 scrambled connectomes. When comparing the true β -value (ADNI: $\beta = 0.757$, BioFINDER, $\beta = 0.603$) with the distribution of null model β -values (ADNI: mean/SD=0/0.07, BioFINDER: mean/SD=0.05/0.05) using an exact test, we again obtain a p-value < 0.001 . Lastly, we repeated the connectivity-related prediction models of tau spreading shown in Figure 7C (tau-weighted functional connectivity) and 7D (tau- and distance-weighted functional connectivity) using the scrambled connectomes. For tau-weighted functional connectivity, comparison of the true β -value (ADNI: $\beta = 0.471$, BioFINDER: $\beta = 0.400$) with the beta-value null distribution (ADNI: mean/SD=0.08, BioFINDER: mean/SD=0.04/0.03) yielded an exact $p < 0.001$. For tau- & distance weighted functional connectivity, comparison of the true β -values (ADNI: $\beta = 0.499$, BioFINDER: $\beta = 0.421$) with the β -value null distribution (ADNI: mean/SD=0.16/0.02, BioFINDER: mean/SD=0.02/0.02) also yielded an exact $p < 0.001$. This suggests that our results are not simply based on the data structure, but on the association between functional connectivity and tau change. We thank the reviewer for this insightful comment and have added these additional analyses to the statistics (p24, p.26&27) and results section (p.6-13) of the manuscript.

2. This may seem like a small point but it is nonetheless important. Applying a single meaningful SUVR "cutoff" to the entire brain is simply not appropriate with AV1451 SUVR data. Other groups have published with global cutoffs, but this was in the early days of AV1451-PET, under assumptions that it behaves the same as other tracers, and in lieu of a better way to describe the data. We now know that the distribution of AV1451 values vary widely by region, and a single SUVR cutoff is therefore inappropriate. This is why I previously encouraged a one-sample t-test, which the authors have applied to the longitudinal data in Figure 1. However, Figure 1 still contains cross-sectional data based on a non-empirical cutoff of 1.3. In order to avoid negatively influencing the field with what will surely be a widely read paper, please either apply a one-sample t-test or some other more empirical threshold to these cross-sectional images, or simply show the data without a threshold.

Response: We appreciate the reviewers comment that AV1451 cut-offs may vary across ROIs. However, we applied a global threshold of 1.3, which was recommended by reviewer 1 and which has been previously used by other authors as a "pathological" tau-PET SUVR cut-off (see for example ¹). Nevertheless, we agree with the reviewer that displaying continuous tau-PET uptake is more appropriate to visualize the distribution of tau-PET uptake across the entire cortex, independent of any threshold. In order to find a compromise between thresholded and continuous data, we have updated Figure 1 so that continuous tau-PET maps are shown, while highlighting regions that surpass the previously used abnormality threshold of $SUVR > 1.3$. Please note also that we had accidentally applied a lower threshold in the map of the CN- $A\beta^+$ of the ADNI cohort in the previous version of the manuscript, which is corrected in the updated figure.

3. As a minor point, the authors may want to mention in the limitations that the use of only a single MRI for two time points of data may expose them to the asymmetry bias (Reuter & Fischl, 2011, Neuroimage).

Response: Thanks, we have now added a sentence to the discussion highlighting this potential limitation (p17-18).

References:

1. Jack CR, *et al.* The bivariate distribution of amyloid-beta and tau: relationship with established neurocognitive clinical syndromes. *Brain : a journal of neurology*, (2019).

Reviewers' comments:

Reviewer #3 (Remarks to the Author):

The authors have addressed all of my concerns. This is an excellent paper. Congratulations!

--Jake Vogel

Reviewer #4 (Remarks to the Author):

Previous review rounds have substantially improved this exciting manuscript, showing that functional connectivity is associated with accumulation of tau pathology. The fact that this can be replicated in an independent set is impressive. The manuscript is very well written.

I have a few remaining issues:

1. How can the authors assure that the size of these 400 ROIs (or even 200) are within the spatial resolution or PSF of the PET data? I went back to the previous manuscript of the authors (Brain) and could not see whether they changed the reconstruction of the PET data or how they dealt with this issue. Partial volume correction may help to some extent, but is limited for smaller regions, especially considering the folding pattern of the cortex.
2. Page 7: the authors state that the association between functional connectivity and tau accumulation was assessed only within the amyloid+ group, because the amyloid- group exhibited no strong increases in tau on a group level. However, this is not the case for the BIOFINDER study and it would be valuable to assess group differences to understand the contribution of amyloidosis.
3. Have the others performed their main analyses (relationship between functional network connectivity and tau change in amyloid+) performed while controlling for diagnosis or MMSE score (instead of APOE), considering the fact that the amyloid+ group consist of both controls and MCI patients (and AD in the BIOFINDER).

Minor issues:

- The term abnormal is not ideal, it suggests that below the cut-off would be normal (no tau), while the PET has detection limits and it is likely that these lower levels of tau can be harmful. Furthermore, threshold are chosen arbitrarily and often sample dependent. I would encourage the authors to use terms as "elevated" versus "low"
- Replace the word "gender" by sex to make sure you are referring to a biological construct.
- The term "predict" should preferably not be used, as these are associations
- Please include the proportion of APOE E4 carriage for each group in Table 1

Reviewers' comments:

Reviewer #3 (Remarks to the Author):

The authors have addressed all of my concerns. This is an excellent paper. Congratulations!

--Jake Vogel

Response: Thanks for these encouraging remarks Jake!

Reviewer #4 (Remarks to the Author):

Previous review rounds have substantially improved this exciting manuscript, showing that functional connectivity is associated with accumulation of tau pathology. The fact that this can be replicated in an independent set is impressive. The manuscript is very well written. I have a few remaining issues:

1. How can the authors assure that the size of these 400 ROIs (or even 200) are within the spatial resolution or PSF of the PET data? I went back to the previous manuscript of the authors (Brain) and could not see whether they changed the reconstruction of the PET data or how they dealt with this issue. Partial volume correction may help to some extent, but is limited for smaller regions, especially considering the folding pattern of the cortex.

Response: The reviewer raises an important point, namely whether the relatively low spatial resolution of the tau-PET images and the different cortical folding pattern across subjects may influence our results. First, to minimize differences in the cortical folding pattern across subjects, we applied structural MRI-based high-dimensional spatial normalization to the co-registered tau-PET images. Second, all tau-PET images in ADNI are sampled by the ADNI-PET core to a uniform spatial resolution of 8mm, which corresponds to the approximate resolution of the lowest resolution scanners that are used in ADNI. For BioFINDER the spatial resolution of the tau-PET images corresponds approximately to 5mm. According to these scanning resolutions, volumes of 8^3 i.e. 512mm^3 can be captured in ADNI vs. 5^3 i.e. 125mm^3 in BioFINDER. The ROI volume ranges from 448mm^3 to 7184mm^3 in the 400-ROI parcellation vs. 912mm^3 to 11568mm^3 in the 200 ROI parcellation. Thus, the volume of only one ROI of the 400 ROI parcellation falls below the minimum volume of 512mm^3 that can be captured by the spatial resolution of the images in ADNI. However, the used ROIs are not strictly cubic in shape and include several voxels, which will lead to blurring of ROI borders during the PET acquisition. However, if autocorrelations due to insufficient spatial resolution would drive the correlation between FC and covariance in tau PET change, Euclidian distance between the ROIs should be a major predictor of that correlation. However, we reported an association between functional connectivity and covariance in tau PET change even after controlling for Euclidian distance, where Euclidian distance itself was a much poorer predictor of covariance in tau change compared to FC. Still, in order to further test whether ROI size had an effect on our results, we performed sensitivity analyses by splitting the 400 ROIs into quartiles based on their size (i.e. number of voxels within an ROI). Specifically, we stratified the ROIs into quartiles ranging from the largest size (i.e. 4th quartile) to the smallest size (i.e. 1st quartile). Within each quartile of high (4th quartile) to low ROI size (1st quartile), we then tested the association between functional connectivity and covariance in tau change. For the 4th quartile including the largest ROIs, we found an association of $b=0.44$, $p<0.001$ in ADNI and $b=0.41$, $p<0.001$ in BioFINDER. For the 3rd quartile, we found an association of $b=0.42$, $p<0.001$ in ADNI and $b=0.37$, $p<0.001$ in BioFINDER. For the 2nd quartile we found an association of $b=0.38$, $p<0.001$ in ADNI and $b=0.28$, $p<0.001$ in BioFINDER. For the 1st quartile including the smallest ROIs we found an association of $b=0.41$, $p<0.001$ in ADNI and $b=0.24$, $p<0.001$ in BioFINDER. A consistent result pattern was also found when conducting this sensitivity analysis using the 200 ROI parcellation (4th quartile, ADNI, $b=0.46$, $p<0.001$; BioFINDER: $b=0.42$, $p<0.001$; 3rd quartile: ADNI: $b=0.36$, $p<0.001$; BioFINDER: $b=0.37$, $p<0.001$; 2nd quartile: ADNI: $b=0.36$, $p<0.001$; BioFINDER: $p=0.30$, $p<0.001$; 1st quartile: ADNI: $b=0.42$, $p<0.001$; BioFINDER: $b=0.25$, $p<0.001$). Consistent results are also found in BioFINDER when using partial volume corrected data. According to these analyses, there is a robust association between functional connectivity and covariance in tau change across different ROI sizes and brain parcellations. This suggests that the limited spatial resolution of tau-PET imaging and fMRI does not account for the observed association between functional connectivity and covariance in tau PET change. Due to the already complex set of analyses we did not include this sensitivity analysis in the main manuscript, however, it will be available to the interested reader in the uploaded rebuttal letter in case of publication.

2. Page 7: the authors state that the association between functional connectivity and tau accumulation was assessed only within the amyloid+ group, because the amyloid- group exhibited no strong increases in tau on a group level. However, this is not the case for the BIOFINDER study and it would be valuable to assess group differences to understand the contribution of amyloidosis.

Response: The reviewer is correct that there is some statistically significant group-average tau-PET accumulation found in the amyloid-negative group of the BioFINDER cohort, potentially due to the longer follow-up duration. Despite statistical significance, these tau increases are weak compared to the amyloid-positive group and do not surpass the abnormality threshold of 1.3 as can be seen the unthresholded group-mean tau-PET images

displayed in Figure 1. However, to address the reviewers request, we computed covariance in tau-PET change matrices in this amyloid-negative group of the BioFINDER cohort and tested the association with functional connectivity. As expected from the overall little tau change in this group, we find a very low, but significant positive association ($b=0.1$, $p<0.001$, see Figure below), which remains after controlling for Euclidean distance ($b=0.09$, $p<0.001$). Congruent associations are found when this analysis is conducted stratified for each functional network (Visual: $b=0.15$, $p<0.001$; Motor: $b=0.14$, $p<0.001$; Dorsal Attention: $b=0.13$, $p<0.001$; Ventral Attention: $b=0.24$, $p<0.001$; Limbic= 0.38 , $p<0.001$; Fronto-parietal Control: $b=0.12$, $p<0.001$; Default-Mode: $b=0.11$, $p<0.001$). Also, the mean levels of covariance in tau change are on average much lower in the amyloid negative (mean/SD= $0.213/0.297$) than in the amyloid positive group (mean/SD= $0.445/0.191$), supporting the fact that there is in general rather low tau and covariance in tau change in the amyloid-negative subjects. We have added this exploratory analyses to the results section of the manuscript (p.8-9).

3. Have the others performed their main analyses (relationship between functional network connectivity and tau change in amyloid+) performed while controlling for diagnosis or MMSE score (instead of APOE), considering the fact that the amyloid+ group consist of both controls and MCI patients (and AD in the BIOFINDER).

Response: This is an important comment. To address the reviewers concern, we have computed residualized change in tau covariance matrices, this time also regressing out effects of diagnosis and MMSE (in addition to sex, education, age and ApoE). In line with our previous analyses controlling for age, sex, education and ApoE4 status, we find congruent associations in the amyloid-positive groups between functional connectivity and residualized covariance in tau-PET change in ADNI ($b=0.38$, $p<0.001$) and BioFINDER ($b=0.24$, $p<0.001$). We have updated the relevant analyses in the results and methods section of the manuscript (p.8 & 26).

Minor issues:

- The term abnormal is not ideal, it suggests that below the cut-off would be normal (no tau), while the PET has detection limits and it is likely that these lower levels of tau can be harmful. Furthermore, threshold are chosen arbitrarily and often sample dependent. I would encourage the authors to use terms as “elevated” versus “low”

Response: We agree with the reviewer and have changed our terminology to “elevated” vs. “low” throughout the manuscript.

- Replace the word “gender” by sex to make sure you are referring to a biological construct.

Response: Following the reviewers comment we have replaced the term gender by sex, to emphasize that we refer to a biological construct.

- The term “predict” should preferably not be used, as these are associations

Response: We agree with the reviewer that the term “predict” should not be used to describe associative results. Thus, we have changed our terminology when describing purely associative results (e.g. association between functional connectivity and covariance in tau change). However, we kept the term “predictor” when referring to the setup of linear regression models including baseline connectivity and baseline tau-PET data to predict future tau changes at the group and individual level.

- Please include the proportion of APOE E4 carriage for each group in Table 1

Response: The proportion of ApoE e4 carriers is now shown in Table 1

REVIEWERS' COMMENTS:

Reviewer #4 (Remarks to the Author):

I thank the authors for the detailed explanation and additional analyses. This is a great manuscript.

There is one minor comment regarding my second concern: the authors now provide the results within the amyloid negative group, but should include a formal testing of comparing amyloid negative versus amyloid positive groups (given that they claim a stronger effect in amyloid positive than amyloid negatives in the second sentence of the discussion and similar inferences are made further down in the discussion).

Reviewer #4 (Remarks to the Author):

I thank the authors for the detailed explanation and additional analyses. This is a great manuscript. There is one minor comment regarding my second concern: the authors now provide the results within the amyloid negative group, but should include a formal testing of comparing amyloid negative versus amyloid positive groups (given that they claim a stronger effect in amyloid positive than amyloid negatives in the second sentence of the discussion and similar inferences are made further down in the discussion).

Response: We thank the reviewer for these encouraging remarks. To address the reviewers' final concern, we included a formal test to determine whether the association between functional connectivity and covariance in tau change was higher in amyloid positive than in amyloid negative subjects within the BioFINDER sample. To this end, we made use of a modified bootstrapping approach: Specifically, we iteratively determined the covariance in tau change within each group using 1000 iterations, within each of which we randomly sampled subjects (with replacement) from the underlying pool of amyloid-positive and amyloid-negative subjects, respectively. Then, we tested for each bootstrapping iteration and group the association between covariance in tau change and functional connectivity. This analysis yielded for each group (i.e. amyloid-positive and negative) a distribution of 1000 beta-values on the association between functional connectivity and bootstrapped covariance in tau change. For the beta-value distributions, we found a mean/SD of 0.22/0.03 for amyloid-positive and 0.06/0.02 for amyloid-negative groups, which were significantly different from each other, as indicated by a two-sample t-test ($p < 0.001$). This additional analysis supports the notion that the association between functional connectivity and covariance in tau change is higher in amyloid-positive than in amyloid-negative subjects. We have added these exploratory analyses to the results section of the manuscript (p.9).